

# 1 A new bed elevation model for the Weddell Sea sector of the West
# 2 Antarctic Ice Sheet

Hafeez Jeofry[1,2], Neil Ross[3], Hugh F.J. Corr[4], Jilu Li[5], Prasad Gogineni[6] and Martin J. Siegert[1]
[1] Grantham Institute and Department of Earth Science and Engineering, Imperial College London,
South Kensington, London, UK
[2] School of Marine Science and Environment, Universiti Malaysia Terengganu, Kuala Terengganu,
Terengganu, Malaysia
[3] School of Geography, Politics and Sociology, Newcastle University, Claremont Road, Newcastle
Upon Tyne, UK
[4] British Antarctic Survey, Natural Environment Research Council, Cambridge, UK
[5] Center for the Remote Sensing of Ice Sheets, University of Kansas, Lawrence, Kansas, USA
[6] ECE Department, The University of Alabama, Tuscaloosa, AL 35487, USA
*Correspondence to*: Hafeez Jeofry (h.jeofry15@imperial.ac.uk)
**Abstract.** We present a new bed elevation digital elevation model (DEM), with a 1 km spatial resolution, for the Weddell Sea
sector of the West Antarctic Ice Sheet. The DEM has a total area of ~125,000 km$^2$ covering the Institute, Möller and Foundation
ice streams and the Bungenstock ice rise. In comparison with the Bedmap2 product, our DEM includes several new
aerogeophysical datasets acquired by the Center for Remote Sensing of Ice Sheets (CReSIS) through the NASA Operation
IceBridge (OIB) program in 2012, 2014 and 2016. We also update bed elevation information from the single largest existing
dataset in the region, collected by the British Antarctic Survey (BAS) Polarimetric Airborne Survey Instrument (PASIN) in
2010-11, as BEDMAP2 included only relatively crude ice thickness measurements determined in the field for quality control
purposes. This have resulted in the deep parts of the topography not being visible in the fieldwork non-SAR processed
radargrams. While the gross form of the new DEM is similar to Bedmap2, there are some notable differences. For example,
the position and size of a deep trough (~2 km below sea level) between the ice sheet interior and the grounding line of
Foundation ice stream has been redefined. From the revised DEM, we are able to better derive the expected routing of basal
water at the ice-bed interface, and by comparison with that calculated using Bedmap2 we are able to assess regions where
hydraulic flow is sensitive to change. Given the sensitivity of this sector of the ice sheet to ocean-induced melting at the
grounding line, especially in light of improved definition of the Foundation ice stream trough, our revised DEM will be of
value to ice-sheet modelling in efforts to quantify future glaciological changes in the region, and therefore the potential impact
on global sea level. The new 1 km bed elevation product of the Weddell Sea sector, West Antarctica can be found in the
http://doi.org/10.5281/zenodo.1035488.

## 35 1. Introduction

The Intergovernmental Panel on Climate Change (IPCC) projected that global sea level could increase by between 0.26 and
0.82 m by the end of the 21st century (Stocker, 2014). The rising oceans pose a threat to the socio-economic activities of
hundreds of millions of people, mostly in Asia, living at and close to the coastal environment. Several processes drive sea level
rise (e.g. thermal expansion of the oceans), but the largest potential factor comes from the ice sheets in Antarctica. The West
Antarctic Ice Sheet (WAIS), which if melted would raise sea-level by around 3.5 m, is grounded on a bed which is ~2 km
below sea level (Bamber et al., 2009b; Ross et al., 2012; Fretwell et al., 2013) allowing the ice margin to have direct contact
with ocean water. One of the most sensitive regions of the WAIS to potential future ocean warming is the Weddell Sea (WS)
sector (Ross et al., 2012; Wright et al., 2014). Ocean modelling studies project that changes in present ocean circulation could
bring warm ocean water into direct contact with the grounding lines at the base of the Filchner Ronne Ice Shelf (FRIS) (Hellmer
et al., 2012; Wright et al., 2014; Martin et al., 2015; Ritz et al., 2015; Thoma et al., 2015), which would act in a similar manner
to the ocean-induced basal melting under the Pine Island Glacier ice shelf (Jacobs et al., 2011). Enhanced melting of the FRIS
could lead to a decrease in the buttressing support to the upstream grounded ice, encouraging enhanced flow from the parent
ice mass to the ocean (Jacobs et al., 2011). A recent modelling study, using a general ocean circulation model coupled together
with a 3D thermodynamic ice sheet model, simulated the inflow of warm ocean water into the Ronne-Filchner ice shelf cavity
on a 1000 year timescale (Thoma et al., 2015). A second modelling study, this time using an ice sheet model, indicated that
the Institute and Möller ice streams are highly sensitive to melting at the grounding lines; with grounding line retreat up to 180
km possible across the Institute and Möller ice streams (Wright et al., 2014). While the Foundation ice stream was shown to



be relatively resistant to ocean-induced change (Wright et al., 2014), a dearth of geophysical measurements of ice thickness
across the ice stream at the time means the result may be inaccurate.
Recently, investigations of ice-sheet change have been regarded as a scientific priority for the next two decades and
beyond (Kennicutt et al., 2014; Kennicutt et al., 2016). While satellite images can provide evidence of surface elevation
changes, they are unable to provide information on the subglacial environment and, hence, are limited in their ability to uncover
ice-sheet flow processes and their change.
The primary tool for measurements of subglacial topography and basal ice-sheet conditions is radio-echo sounding (RES)
(Dowdeswell and Evans, 2004; Bingham and Siegert, 2007).The first topographic representation of the land surface beneath
the entire Antarctic ice sheet (Drewry, 1983) was published by the Scott Polar Research Institute, University of Cambridge,
(SPRI) in collaboration with the US National Science Foundation Division of Polar Programmes (NSF-DPP) and the Technical
University of Denmark (TUD), following multiple field seasons of RES surveying in the 1970s (Drewry and Meldrum, 1978;
Drewry et al., 1980; Jankowski and Drewry, 1981; Drewry, 1983). The compilation (Drewry, 1983) included folio maps of
bed topography and ice-sheet surface elevation and thickness. The bed was digitized on a 20 km grid for use in ice-sheet
modelling (Budd et al., 1984). However, only around one third of the continent was measured at a line spacing of less than
~100 km (Lythe and Vaughan, 2001), making the elevation product highly erroneous in many places, with obvious knock-on
consequences for modelling. Several RES campaigns were thenceforth conducted, and the data from them were compiled into
a single new Antarctic bed elevation product, named Bedmap (Lythe and Vaughan, 2001). The Bedmap DEM had a cell
resolution of 5 km, and included over 1.4 million km and 250,000 km line track of airborne and ground-based radio-echo
sounding data, respectively. Subglacial topography was extended north to $60^0$S, for purposes of ice sheet modelling, and
determination of ice-ocean interactions. Since its release, Bedmap has proved to be highly useful for a wide range of research,
yet inherent errors within it (e.g. inaccuracies in the DEM and conflicting grounding line representation with observations)
limited its effectiveness to the community (Le Brocq et al., 2010; Fretwell et al., 2013). After 2001, several new RES surveys
were conducted, to fill data gaps revealed by Bedmap, especially during and after the 4th International Polar Year (2007-9).
These new data led to the most recent Antarctic bed project, named Bedmap2 (Fretwell et al., 2013). Despite significant
improvements in the resolution and accuracy of Bedmap2 compared with Bedmap, a number of inaccuracies and poorly
sampled areas persist where only 34% of cells have data and 80% have data within the range of 20 km (Fretwell et al., 2013;
Pritchard, 2014), preventing a comprehensive appreciation of the complex relation between the topography and internal ice-
sheet processes, and indeed a full appreciation of the sensitivity of the Antarctic ice sheet to ocean and atmospheric warming.
The Weddell Sea sector of the WAIS was the subject of a major aerogeophysical survey in 2010-11 (Ross et al., 2012),
revealing the ~2 km deep Robin Subglacial Basin immediately upstream of present-day grounding lines, from which
confirmation of the ice-sheet sensitivity from ice-sheet modelling was determined (Wright et al., 2014). Further geophysical
surveying of the region has been undertaken since Bedmap2 (Siegert et al., 2016b), improving knowledge of the land surface
beneath the ice sheet. This has provided an enhanced appreciation of the importance of basal hydrology to ice flow (Siegert et
al., 2016b), and the complexities of the interaction of basal water flow, bed topography and ice surface elevation, emphasising
the importance of developing accurate and high resolution DEMs (Lindbäck et al., 2014; Siegert et al., 2014; Graham et al.,
89 2016).

In this paper, we present a revised bed DEM for the Weddell Sea sector, based on a compilation of airborne radar surveys.
The DEM has a total area of ~125,000 km$^2$ and a grid cell resolution of 1 km. From this dataset, we reveal the expected
pathways of subglacial water, and discuss the differences between the new DEM and Bedmap2.
**2. Study area**
The study area is located in the Weddell Sea sector of WAIS covering the Institute, Möller and Foundation ice streams, and
the Bungenstock ice rise (Fig. 1). The DEM extends 135 km grid south of the Bungenstock ice rise, 195 km grid east of the
Foundation ice stream, over the Pensacola Mountain and 185 km grid west of the Institute ice stream. In comparison with the
Bedmap2, our new DEM benefits from several new airborne geophysical datasets (e.g. NASA OIB 2012, 2014 and 2016). In
addition, the new DEM is improved by the inclusion of ice thickness picks derived from SAR-processed RES data from the
BAS aerogeophysical survey of the Institute and Möller ice streams conducted in 2010/2011. This has improved the accuracy
of the determination of the ice bed interface in comparison to Bedmap2. The data and methods used in this study are mostly
similar to Jeofry et al. (2017), however, we have expanded the boundaries of our new bed elevation DEM by focussing across
the Weddell Sea sector relative to the Institute ice stream as the main focus in Jeofry et al. (2017). In addition, this paper
focuses on discussing the geomorphology and improvement that have been made between our new bed elevation DEM and
Bedmap2 product whereas Jeofry et al. (2017) discuss on the hitherto unknown subglacial embayment and its effect to the ice-
sheet dynamics.
**3. Data and methods**



The radio echo sounding (RES) data used in this study were compiled from four main sources (Fig. 1a). First, SPRI data
collected during six survey campaigns between 1969 and 1979 (Drewry, 1983). Second, the BAS airborne radar survey
accomplished during the austral summer 2006/07 (GRADES/IMAGE) (Ashmore et al., 2014). Third, the BAS survey of the
Institute and Möller ice streams, undertaken in 2010/2011 (IMAFI) (Ross et al., 2012). Finally, flights conducted by the Center
for the Remote Sensing of Ice Sheet (CReSIS) during the NASA OIB programme in 2012, 2014 and 2016 (Gogineni, 2012)
supplement the data previously used in the Bedmap2 bed elevation product to accurately characterize the subglacial topography
of this part of the Weddell Sea sector.
3.1   Scott Polar Research Institute survey
The SPRI surveys of the 1970s covered a total area of 6.96 million km$^2$ (~40% of continental area) across West and East
Antarctica (Drewry et al., 1980). The data were collected using a pulsed radar system operating at centre frequencies of 60 and
300 MHz (Christensen, 1970; Skou and Søndergaard, 1976) equipped on NSF C-130R Hercules aircraft (Drewry and
Meldrum, 1978; Drewry et al., 1980; Jankowski and Drewry, 1981; Drewry, 1983). The 60 MHz antennas, built by the
Technical University of Denmark, comprised an array of four half wave dipoles which were mounted in neutral aerofoil
architecture of insulating components beneath the starboard wing. The 300 MHz antennas were composed of four dipoles
attached underneath a reflector panel below the port wing. The purpose of this unique design was to improve the backscatter
acquisition and directivity. The returned signals were archived on 35 mm film and dry-silver paper by a fibre optic oscillograph.
Aircraft navigation was assisted using an LTN-51 inertial navigation system giving a positional error of around 3 km.
Navigation and other flight data were stored on magnetic and analogue tape by an Airborne Research Data System (ARDS)
constructed by US Naval Weapons Center. The system recorded up to 100 channels of six-digit data with a sampling rate of
303 Hz per channel. Navigation, ice thickness and ice surface elevation records were recorded every 20 seconds, corresponding
to around 1.6 km between each data point included on the Bedmap2 product. The data was initially recorded on a 35-mm
photographic film (i.e. Z-scope radargrams) and was later scanned and digitized, as part of a NERC Centre for Polar
Observation and Modelling (CPOM), in 2004. Each film records were scanned separately and reformatted to form a single
electronic image of a RES transect. The scanned image was loaded into an image analysis package (i.e. ERDAS Imagine) to
trace the internal layers which were then digitized. The dataset was later standardized with respect to the ice surface.
3.2   BAS GRADES/IMAGE surveys
The GRADES/IMAGE project was conducted during the austral summer of 2006/2007 and consisted of ~27,550 km of RES
data flown across the Antarctic Peninsula, Ellsworth Mountain and Filchner-Ronne ice shelf. The BAS Polarimetric radar
Airborne Science Instrument (PASIN) radar operates at a centre frequency of 150 MHz, has a 10 MHz bandwidth and a pulse-
coded waveform acquisition rate of 312.5 Hz (Corr et al., 2007; Ashmore et al., 2014). The PASIN system interleaves a pulse
and CHIRP signal to acquire two datasets simultaneously. Pulse data is used for imaging layering in the upper half of the ice
column, whilst the more powerful CHIRP is used for imaging the deep-ice and sounding the ice sheet bed. The peak transmitted
power of the system is 4 kW.  The spatial sampling interval of ~20 m resulted in ~50,000 traces of data for a typical 4.5-hour
flight. The radar system consisted of 8 folded dipole elements; 4 transmitters on the port side and 4 receivers on the starboard
side. The receiving "backscatter" signal was digitized and sampled using a sub-Nyquist sampling technique and are then
recorded on magnetic tape drives. The pulses are compressed using a matched filter, and sidelobes are minimized using a
Blackman window. Aircraft position was recorded by an onboard carrier wave global positioning system (GPS). The absolute
positional accuracy for GRADES/IMAGE was 0.1 m (Corr et al., 2007). Synthetic Aperture Radar (SAR) processing was not
applied to the GRADES/IMAGE data.
3.3   BAS Institute-Möller Antarctic Funding Initiative surveys
The BAS data acquired during the IMAFI project consist of ~25,000 km of aerogeophysical data collected during 27 flights
from two field camps. 17 flights were flown from C110 (-82° 37' 30"N, 280° 59' 13.2"E), and the remaining 10 flights were
flown from Patriot Hills (-80° 19' 60"N, 278° 34' 60"E) (Fig. 1b). Data were acquired with the same PASIN radar used for the
GRADES/IMAGE survey. The data rate of 13 Hz gave a spatial sampling interval of ~10 m for IMAFI. The system was
installed on the BAS de Havilland Twin Otter aircraft with a four-element folded dipole array mounted below the starboard
wing used for reception and the identical array attached below the port wing for transmission. The flights were flown in a
stepped pattern during the IMAFI survey to optimise potential field data (gravity and magnetics) acquisition (Fig. 1a). Leica
500 and Novatel DL-V3 GPS receivers were installed in the aircraft, corrected with two Leica 500 GPS base stations which
were operated throughout the survey to calculate the position of the aircraft (Jordan et al., 2013). The positional data were
referenced to the WGS 84 ellipsoid. The absolute positional accuracy for IMAFI (the standard deviation for the GPS positional





error) was calculated to be at 7 cm and 20 cm in the horizontal and vertical dimensions (Jordan et al., 2013). 2-D focused SAR
processing (Hélière et al., 2007) (see section 4.1) was applied to the IMAFI data.
3.4   NASA OIB / CReSIS surveys
The OIB project surveyed a total distance of ~32,693 km, ~52460 km and ~53672 km in Antarctica in 2012, 2014 and 2016,
respectively, using the Multichannel Coherent Radar Depth Sounder (MCoRDS) system developed at the University of Kansas
(Gogineni, 2012). The system was operated with a carrier frequency of 195 MHz, a bandwidth of 10 MHz and 50 MHz in
2012 (Rodriguez-Morales et al., 2014) and 2014 onwards (Siegert et al., 2016b). The radar consisted of a five – element
antenna array housed in a customized antenna fairing which is attached beneath the NASA DC – 8 aircraft fuselage (Rodriguez-
Morales et al., 2014). The five antennas were operated from a multichannel digital direct synthesis (DDS) controlled waveform
generator enabling the user to adjust the frequency, timing, amplitude and phase of each transmitted waveform (Shi et al.,
2010). The radar employs an eight-channel waveform generator to emit eight independent transmit chirp pulses. The system
is capable of supporting 5 receiver channels with Analog Devices AD9640 14 bit analog-to-digital converter (ADC) acquiring
the waveform at a rate of 111 MHz in 2012 (Gogineni, 2012). The system was upgraded in 2014 and 2016 utilizing six channel
chirp generation and supports six receiver channels with a waveform acquisition rate at 150 MHz. Multiple receivers allow
array processing to suppress surface clutter in the cross track direction which could potentially conceal weak echoes from the
ice – bed interface (Rodriguez-Morales et al., 2014). The radar data are synchronized with the GPS and inertial navigation
system (INS) using the GPS time stamp to determine the location of data acquisition.
**4.  Data processing**
Airborne RES assumes the radar pulse propagates through ice at a constant wave speed of 0.168 m ns$^{-1}$, since glacial ice is
assumed to be homogenous (Lythe and Vaughan, 2001). The radar pulse travels through a medium until it meets a boundary
of differing dielectric constant, which causes some of the radio wave to be reflected and subsequently captured by the receiver
antenna. There are three factors causing dielectric contrast in ice sheets; density variations in the upper 700 m, acidic layers
caused by the aerosol from volcanic activity, and ice permittivity variations from crystal fabrics (Siegert, 1999; Corr et al.,
2007). The time travelled by the radar pulse between the upper and lower reflecting surface is measured and converted to ice
thickness with reference to WGS 84 (Fig. 2). The digitized SPRI-NSF-TUD bed picks data are available through the BAS
webpage  (https://data.bas.ac.uk/metadata.php?id=GB/NERC/BAS/AEDC/00326).  The  two-dimensional  SAR-processed
radargrams in SEG-Y format for the IMAFI survey are provided at doi.org/10.5285/8a975b9e-f18c-4c51-9bdb-b00b82da52b8,
whereas the ice thickness datasets in Comma Separated Value (CSV) format for both GRADES/IMAGE and IMAFI are
available via the BAS aerogeophysical processing portal (https://secure.antarctica.ac.uk/data/aerogeo/). The ice thickness data
for IMAFI are provided in two folders; (1) the region of thinner ice (200m) picked from the pulse dataset and (2) the overall
ice thickness data, derived from picking of SAR-processed CHIRP radargrams. The data are arranged according to the latitude,
longitude, ice thickness values and the pulse repetition interval (PRI) radar shot number which is used to index the raw data.
The OIB SAR images (Level 1B) in MAT (binary MATLAB) format and the radar depth sounder Level 2 (L2) data in CSV
format are available via the CReSIS website (https://data.cresis.ku.edu/). The L2 data include measurements for GPS time
during data collection, latitude, longitude, elevation, surface, bottom and thickness. For more information on these data, the
reader  should  refer  to  the  appropriate  CReSIS  data  information  and  guidance  notes  for  each  field  season  (i.e.
https://data.cresis.ku.edu/data/rds/rds_readme.pdf).
4.1   GRADES/IMAGE and Institute-Möller Antarctic Funding Initiative data processing
The waveform was retrieved and sequenced according to its respective transmit pulse type. The modified data were then
collated using Matlab standard format. Doppler filtering (Hélière et al., 2007) was used to remove the backscattering hyperbola
in the along-track direction (Corr et al., 2007; Ross et al., 2012).  Chirp compression was then applied to the along-track data.
Unfocused synthetic aperture (SAR) processing was used for the GRADES/IMAGE survey by applying a moving average of
33 data points (Corr et al., 2007) whereas 2D-SAR (i.e. focused) processing based on Omega-K algorithm was used to process
the IMAFI data (Hélière et al., 2007; Winter et al., 2015) to enhance both along-track resolution and echo signal noise. The
bed echo was depicted in a semi-automatic manner using PROMAX seismic processing software. All picking for IMAFI was
undertaken by a single operator (Neil Ross). A nominal value of 10 m is used to correct for the firn layer during the processing
of ice thickness, which introduces an error of the order of $\pm$ 3 m across the survey field (Ross et al., 2012). This is small relative
to the total error budget of the order of $\pm$ 1%. Finally, the GPS and RES data were combined to determine the ice thickness,
ice-surface and bed elevation datasets. Elevations are measured with reference to WGS84. The ice surface elevation was



calculated by subtracting terrain clearance from the height of the aircraft, whereas the bed elevation was computed by subtracting the ice thickness from the ice surface elevation.

### 4.2 OIB data processing

The OIB radar adopts SAR processing in the along-track direction to provide higher resolution images of the subglacial profile. The data were processed in three steps to improve the signal-to-noise ratio and increase the along-track resolution (Gogineni et al., 2014). The raw data were first converted from a digital quantization level to a receiver voltage level. The surface was captured using the low-gain data, microwave radar or laser altimeter. A normalized matched filter with frequency-domain windowing was then used for pulse compression. 2D-SAR processing was used after conditioning the data, which is based on the frequency-wavenumber (F-K) algorithm. The F-K SAR processing, however, requires a straight and uniformly sampled data which are usually not met in the raw data since the aircraft's speed is not consistent and the trajectory is not straight. The raw data were thus spatially resampled in along-track using sinc kernel to approximate a uniformly sampled dataset. The vertical deviation in aircraft trajectory from the horizontal flight path was compensated in the frequency domain with a time-delay phase shift. The phase shift was later removed for array processing as it is able to account for the non-uniform sampling; the purpose is to maintain the original geometry for the array processing. Array processing was performed in the cross-track flightpath to reduce surface clutter as well as improving the signal-to-noise ratio. Both the delay-and-sum and Minimum Variance Distortionless Response (MVDR) beamformer were used to combine the multichannel data, and for regions with significant surface clutter the MVDR beamformer could effectively minimize the clutter power and pass the desired signal with optimum weights (Harry and Trees, 2002).

### 4.3 Quantifying ice thickness, bed topography and subglacial water flow

The new ice thickness DEM was formed from all the available RES data using the 'Topo to Raster' function in ArcGIS, based on the Australian National University Digital Elevation Model (ANUDEM) elevation gridding algorithm (Hutchinson, 1988). The ice thickness picks from the geophysical data were gridded using the Nearest Neighbour interpolation within Topo to Raster. The ice thickness DEM was then subtracted from the ice-sheet surface elevation derived from the combined European Remote Sensing Satellite-1 (ERS-1) radar and Ice, Cloud and land Elevation Satellite (ICESat) laser satellite altimetry DEM (Bamber et al., 2009a), to derive the bed topography. The ice thickness, ice-sheet surface and bed elevations were then gridded at a uniform 1 km spacing, and referenced to the Polar Stereographic projection (Snyder, 1987) to form the new DEMs. The difference map between the new DEM and the Bedmap2 product (Fretwell et al., 2013) was computed by subtracting the Bedmap2 bed elevation DEM from the new bed elevation DEM. Crossover analysis for the 2006/7 data onwards (including data acquired on flightlines beyond the extent of our DEM) shows the RMS errors of 9.1 m (GRADES/IMAGE), 15.82 m (IMAFI), 45.93 m (CRESIS 2012), 23.74 m (CRESIS 2014) and 20.27 m (CRESIS 2016).

Subglacial water flowpaths were calculated based on the hydraulic potentiometric surface principle, in which basal water pressure is balanced by the ice overburden pressure as follows:

$$\varphi = (\rho_w \times g \times y) + (\rho_i \times g \times h) \tag{1}$$

where $\varphi$ is the theoretical hydropotential surface, $y$ is the bed elevation, $h$ is the ice thickness, $\rho_w$ and $\rho_i$ are the density of water (1000 kg m$^{-3}$) and ice (920 kg m$^{-3}$), assuming ice to be homogenous, respectively, and $g$ is the gravitational constant (9.81 ms$^{-2}$) (Shreve, 1972). Sinks in the hydrostatic pressure field raster were filled to produce realistic hydrologic flowpaths. The 'flow direction' of the raster was then defined by assigning each cell a direction to the steepest downslope neighbouring cell. Sub-basins less than 200 km$^2$ were removed due to the coarse input of bed topography and ice thickness DEMs.

## 5. Results

### 5.1 A new 1-km digital elevation model of the Weddell Sea sector

We present a new DEM of the WS sector of West Antarctica, compiled from several airborne geophysical radar surveys (Fig. 3a). The Bedmap2 bed elevation and the difference map are shown in Fig. 3b and 3c respectively. The new DEM contains substantial changes in certain regions, whereas in others there are consistencies between the two DEMs, for example across the Bungenstock ice rise, where there are little new data. The mean error between the two DEMS is -86.45 m indicating a slightly lower bed elevation in the new DEM data compared to Bedmap2, which is likely the result of deep parts of the topography (i.e. valley bottoms) not being visible in the fieldwork non-SAR processed QC radargrams for the IMAFI project (e.g. Horsehoe Valley, near Patriot Hills in the Ellsworth Mountains (Winter, 2016). The bed elevation measurement upstream



of the Bungenstock ice rise and across the Robin Subglacial Basin shows a generally good agreement with Bedmap2; only some portions of new DEM across the Möller ice stream, Pagano Shear zone and Pirrit Hills area are significantly different from Bedmap2, with differences in bed elevation typically ranging between -109 m to 172 m (Fig. 3c). There is, however, large disagreement between the two DEMs in the western region of Institute ice stream, across the Ellsworth Mountains (e.g. in the Horseshoe valley), the Foundation ice stream and towards East Antarctica where topography is more rugged. It is also worth noting the significant depth of the bed topography beneath the trunk of the Foundation ice stream, where a trough more than ~2 km deep is located and delineated (Fig. 3d). The trough is ~38 km wide and ~80 km in length, with the deepest section ~2.3 km below sea level. The new DEM shows a significant change in the depiction of Foundation Trough; we have measured it to be ~1 km deeper relative to the Bedmap2 product.

In order to further quantify the differences between Bedmap2 and our new DEM, we present terrain profiles of both DEMs relative to four RES flightlines (Fig. 3c). It is worth noting that the new DEM is much more consistent with the bed elevations from the RES data picks compared to Bedmap2 (Fig. 4a and 4b). The new DEM show a correlation coefficient of 0.96 and 0.91 for Profile A and B, respectively. This is slightly higher compared with the Bedmap2 which is 0.94 (Profile A) and 0.91 (Profile B). Although inaccuracies of the bed elevation persist across the Foundation ice stream for both DEMs, the gross pattern of the bed elevation for the new DEM is much more consistent with the RES transect relative to the Bedmap2 (Fig. 4c and 4d) with correlation coefficient of 0.97 and 0.94 for Profile C and D, respectively. In contrast, the correlation coefficient for the profile produced by Bedmap2 (0.87 for Profile C and 0.82 for Profile D) are significantly reduced.

## 5.2 Hydrology

Computing the potential passageway of subglacial water beneath the ice-sheet is critical for comprehending ice-sheet dynamics (Bell, 2008; Stearns et al., 2008; Siegert et al., 2016a). It is noted that the development of subglacial hydrology pathways is highly sensitive to ice-surface elevation, and to a lesser degree by bed morphology (Wright et al., 2008; Horgan et al., 2013). Figure 3d shows a comparison of subglacial hydrology pathways between our new bed DEM and the Bedmap2 DEM. The gross patterns of water flow are largely unchanged between the two DEMs, especially across and upstream of Institute and Möller ice stream. The similar water pathway pattern between both DEMs in these regions is also consistent with the small error in bed topography (Fig. 3c). Despite large differences in bed topography across the Foundation ice stream and the Ellsworth Mountains region, the large-scale patterns of water flow are also similar between both DEMs; this is due to the dominance of the ice-surface slope in driving basal water flow in these regions (Shreve, 1972). Nonetheless, there are several local small-scale differences in the water pathways (Fig. 3d), which highlight hydraulic sensitivity. The subglacial water network observed in the new DEM across the Foundation ice stream appears to be more arborescent than that derived from Bedmap2. This is due to the introduction of new data, resulting in a better-defined bed across the Foundation Trough (Fig. 3d). It is also worth noting that the pattern of the subglacial water pathway observed in the new DEM adjacent to the grounding line across the Möller ice stream is almost in good agreement with the position of sub-ice-shelf channels, which have been delineated from a combination of satellite images and RES data (Le Brocq et al., 2013).

Subglacial lakes discovered across the WS sector form a vital constituent of the basal hydrological system (Wright and Siegert, 2012). These lakes exist due to sufficient amount of geothermal heating (50 mW m$^{-2}$ – 70 mW m$^{-2}$) which melts the base of the ice sheet. In addition, the pressure exerted by the overlying ice causes the melting point of the ice to be lowered. At the western margin of the Ellsworth Mountain lies a large body of the subglacial water known as Ellsworth Subglacial Lake (-78° 59' 24"N, 269° 25' 48"E). The lake measures 28.9 km$^2$ with a depth ranging between 52 m to 156 m capable of carrying a water body volume of 1.37 km$^3$ (Woodward et al., 2010). There are 4 known subglacial lakes distributed across the Institute ice stream (Wright and Siegert, 2012). The Institute W1 (-81° 24' 3.6"N, 282° 27' 18"E) is located close to the Robin Subglacial Basin A whereas Institute W2 (-81° 37' 40.8"N, 276° 25' 15.6"E) is located to the northeast of Pirrit Hill. The Institute E1 (-82° 7' 48"N, 285° 30' 32.4"E) and E2 (-82° 37' 30"N, 280° 59' 13.2"E) are located to the southwest of Robin Subglacial Basin B and near the field camp of C110, respectively. In addition, there are three subglacial lakes in the Foundation ice stream catchment: Foundation 1 (-84° 31' 19.2"N, 286° 20' 16.8"E); Foundation 2 (-84° 59' 2.4"N, 286° 1' 44.4"E); and Foundation 3 (-85° 15' 28.8"N, 287° 12' 25.2"E) (Wright and Siegert, 2012).

## 5.3 Geomorphological description of the bed topography

The WS sector of WAIS is composed of three major ice-sheet outlets (Fig. 1a); the Institute, Möller and Foundation ice streams, feeding ice to the FRIS, the second largest ice shelf in Antarctica after the Ross Ice Shelf. Recent geophysical inspection of the subglacial topography in the region reveals features such as steep reverse bed slopes, similar in scale to that measured for upstream Thwaites Glacier, close to the Institute and Möller ice stream grounding lines. The bed slopes inland to a ~1.8 km deep basin (the Robin Subglacial Basin), which is divided into two sections, with few obvious significant ice-sheet pinning points (Ross et al., 2012). Elevated beds in other parts of the WS sector allow the ice shelf to ground, causing ice surface




features known as ice rises and rumples (Matsuoka et al., 2015). The Bungenstock ice rise (Siegert et al., 2013) is excellent
example of an ice rise, though peculiarly is an ice rise that is grounded far below sea level.
335        The Institute ice stream has three tributaries, within our survey grid, to the south and west of the Ellsworth Mountains,
occupying the Horseshoe Valley, Independence and Ellsworth troughs (Fig. 1b) (Winter et al., 2015). The Horseshoe Valley
Trough, around 20 km wide and 1.3 km below sea level at its deepest point, is located downstream of the steep mountains of
the Heritage Range. A subglacial ridge is located between the mouth of the Horseshoe Valley Trough and the main trunk of
the Institute ice stream (Winter et al., 2015). The Independence Trough is located subparallel to the Horseshoe Valley Trough,
separated by the 1.4 km high of the Independence Hills. The trough is ~22 km wide and is 1.1 km below sea level at its deepest
point. It is characterized by two distinctive plateaus (~6 km wide each) on each side of the trough, aligned alongside the main
trough axis. Ice flows eastward through the Independence Trough for ~54 km before it shifts to a northward direction where
the trough widens to 50 km and its connection with the main Institute ice stream. The Ellsworth Trough is aligned with the
Independence Trough, and both are orthogonal to the orientation of the Amundsen – Weddell ice divide, dissecting the
Ellsworth Subglacial Highlands northwest to southeast. The Ellsworth Trough measures ~34 km in width and is ~2 km below
sea level at its deepest point, and is ~260 km in length. It is considered to be the largest and deepest trough-controlled tributary
in this region (Winter et al., 2015). The Ellsworth Trough is intersected by several smaller valleys aligned perpendicularly to
the main axis, which are relics landforms from a previous small dynamic ice mass (predating the WAIS in its present
configuration) (Ross et al., 2014). The Ellsworth Trough contains ~15 – 20 km long Ellsworth Subglacial Lake (Siegert et al.,
2004a; Woodward et al., 2010; Siegert et al., 2012; Wright and Siegert, 2012).
351        Satellite altimetry and imagery are able to estimate the grounding line that separates the grounded ice sheet from the
floating ice shelf, based on surface changes due to tidal oscillations and the subtle ice surface features. Such analysis is prone
to uncertainty, however. There are currently four proposed grounding line locations, based on different satellite data sets and/or
methods of analysis (Bohlander and Scambos, 2007; Bindschadler et al., 2011; Brunt et al., 2011; Rignot et al., 2011a). Each
of the grounding lines were delineated from satellite images, but without direct measurement of the subglacial environment.
This results in ambiguities in the precise location of the transition between floating and grounded ice (Jeofry et al., 2017). In
addition, RES data have demonstrated clear errors in the position of the grounding line with a large, hitherto unknown,
subglacial embayment near the Institute ice stream grounding line. The subglacial embayment is ~1 km deep and is potentially
open to the ice shelf cavity, causing the inland ice sheet to have a direct contact with ocean water. In addition, our RES analysis
reveals a better- defined Foundation Trough, in which the grounding line is perched on very deep topography around 2 km
below sea level.
362        A previous study revealed a series of ancient large sub-parallel subglacial bed channels between MIS and FIS, adjacent
to the Marginal Basins (Fig. 1b) (Rose et al., 2014). While these subglacial channels are likely to have been formed by the
flow of basal water, they are presently located beneath slow moving and cold-based ice. It is thought, therefore, that the
channels are ancient and were formed at a time when surface melting was prevalent in West Antarctica (e.g. the Pliocene).
366        The bed topography of the WS appears both rough (over the mountains and exposed bedrock) and smooth (across the
sediment-filled regions) (Bingham and Siegert, 2007). Although there is no specific method or standardized unit to measure
bed roughness, studies of bed roughness calculated using Fast Fourier Transform (FFT) technique based on the relative
measurement of bed obstacle amplitude and frequency of the roughness obstacles have indicated that the IIS and MIS are
dominated by relatively low roughness values, less than 0.1 (Bingham and Siegert, 2007; Rippin et al., 2014), which was
suggested as being the result of the emplacement of marine sediments as in the Siple Coast region (Siegert et al., 2004b; Peters
et al., 2005). Radar-derived roughness analysis has evidenced a smooth bed across the Robin Subglacial Basin where sediments
may exist (Rippin et al., 2014). The deepest parts of the Robin Subglacial Basin are anomalously rough,  marking the edge of
a sedimentary drape where the highest ice flow velocities are generated (Siegert et al., 2016b). As such, the smooth basal
topography of the Institute and Möller ice stream catchments is less extensive than proposed by Bingham and Siegert (2007).
It has been demonstrated since, that the subglacial topography of the region between the Robin Subglacial Basin and the Pirrit
and Martin-Nash Hills is relatively flat, smooth and gently sloping, and has been interpreted as a bedrock planation surface
(Rose et al., 2015). This zone of the study area was originally interpreted by Bingham and Siegert (2009) as underlain by fine-
grained marine sediments, due to the limited RES data available to them at the time. Although the exact formation process of
the planation surface is unknown, it is thought that this geomorphological feature formed due to marine and/or fluvial erosion
(Rose et al., 2015).
**6.   Data availability**
The new 1 km bed elevation product of the Weddell Sea sector, West Antarctica can be found in the http://doi.org/10.5281/zeno
do.1035488. We used four radar datasets to construct the 1 km ice thickness DEM, as follows: (1) Digitized radar data from
the 1970s SPRI-NSF-TUD surveys, in which the bed was picked every 15-20 seconds (1-2 km), recorded here in an Excel 97-
2003 Worksheet (XLS), which can be obtained from the UK Polar Data Centre (UKPDC) website at





https://data.bas.ac.uk/metadata.php?id=GB/NERC/BAS/AEDC/00326; (2) BAS GRADES/IMAGE and (3) BAS IMAFI
airborne surveys, both available from the UKPDC Polar Airborne Geophysics Data Portal at https://secure.antarctica.ac.uk/da
ta/aerogeo/; and (4) NASA Operation IceBridge radar depth sounder Level 2 (L2) data, available from the Center for Remote
Sensing of Ice Sheet (CReSIS) website at https://data.cresis.ku.edu/.
The 1 km ice-sheet surface elevation DEM was derived from a combination of ERS-1 surface radar and ICESat laser altimetry,
which is downloadable from the National Snow and Ice Data Center (NSIDC) website at https://nsidc.org/data/docs/daac/nsidc
0422_antarctic_1km_dem/.
Two-dimensional SAR-processed radargrams in SEG-Y format for the BAS IMAFI airborne survey, and The NASA Operation
IceBridge SAR images (Level 1B) in MAT (binary MATLAB) format, are provided at doi.org/10.5285/8a975b9e-f18c-
4c519bdb-b00b82da52b8 and https://data.cresis.ku.edu/, respectively.
Ancillary information for the MEaSUREs InSAR-based ice velocity map of Central Antarctica can be found at
doi:10.5067/MEASURES/CRYOSPHERE/nsidc-0484.001 and the MODIS Mosaic of Antarctica 2008 – 2009 (MOA2009)
ice sheet surface image map is available at doi.org/10.7265/N5KP8037. The RADARSAT (25m) ice-sheet surface satellite
imagery is accessible from the Byrd Polar and Climate Research Center website at http://research.bpcrc.osu.edu/rsl/radarsat/da
ta/.
A summary of the data used in this paper, and their availability is provided in the table 1.
**7. Conclusions**
We have compiled airborne radar data from a number of geophysical surveys, including the SPRI-NSF-TUD surveys of the
1970s, the GRADES/IMAGE and IMAFI surveys acquired by BAS in 2006/7 and 2010/11, respectively, and new geophysical
datasets collected by the CReSIS from the NASA OIB project in 2012, 2014 and 2016. From these data, we produce a bed
topography DEM with high spatial resolution (1km). The DEM covers a total area of ~125,000 km$^2$ of the Weddell Sea sector
including the Institute ice stream, Bungenstock ice rise, Möller and Foundation ice streams. Large differences can be observed
between the new DEM and that of the previous DEM of the region (Bedmap2), most notably across the Foundation ice stream,
where we reveal the grounding line to be resting on a bed ~2 km below sea level, with a deep trough immediately upstream as
deep as 2.3 km below sea level. In addition, the bed elevation of our DEM appears to be slightly lower relative to the bed
elevation of Bedmap2 DEM, which is likely the results of the deep sections of the topography not being visible in the fieldwork
non-SAR processed radargrams. Our new DEM also revises the pattern of potential basal water flow across the Foundation
ice stream and towards East Antarctica in comparison to that derived from Bedmap2. Our new DEM, and the data used to
compile it, is available to download, and will be of value to ice-sheet modelling experiments in which the accuracy of the DEM
is important to ice flow processes in this particularly sensitive region of the WAIS.
**Author contributions.** Hafeez Jeofry carried out the analysis, created the figures, wrote the paper and compiled the database.
All authors contributed to the database compilation, analysis and writing of the paper.
**Competing interests.** The authors declare that they have no conflict of interest.
**Acknowledgement.** The data used in this project are available at the Center for the Remote Sensing of Ice data portal
https://data.cresis.ku.edu/ and at the UK Airborne Geophysics Data Portal https://secure.antarc-tica.ac.uk/data/aerogeo/.
Prasad Gogineni and Jilu Li acknowledge funding by NASA for CReSIS data collection and development of radars
(NNX10AT68G), Martin J. Siegert, Neil Ross, Hugh F.J. Corr, Fausto Ferraccioli, Rob Bingham, Anne Le Brocq, David
Rippin, Tom Jordan, Carl Robinson, Doug Cochrane, Ian Potten and Mark Oostlander acknowledge funding from the NERC
Antarctic Funding Initiative (NE/G013071/1) and Hafeez Jeofry acknowledges funding from the Ministry of Higher Education
Malaysia and the Norwegian Polar Institute for the Quantarctica GIS package.



**a.**

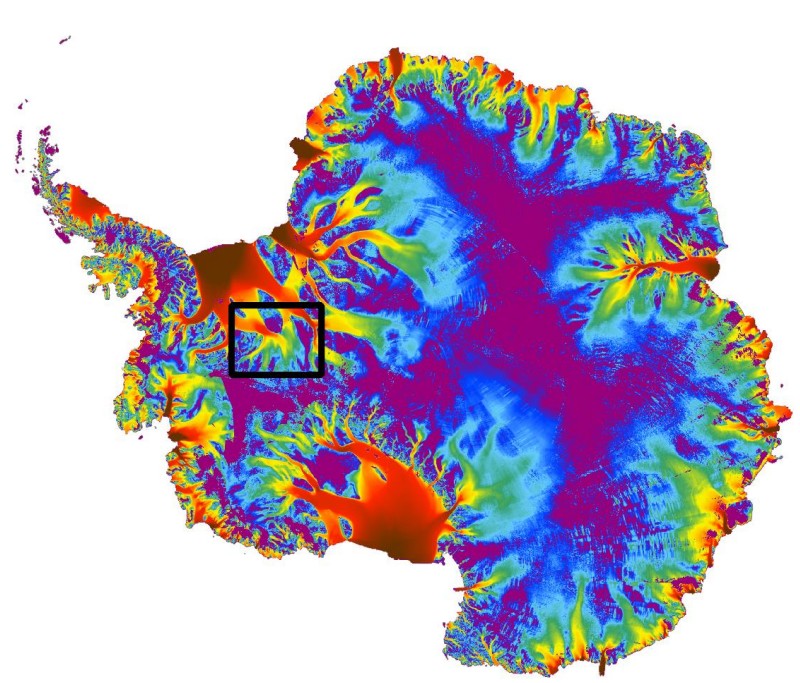

**b.**

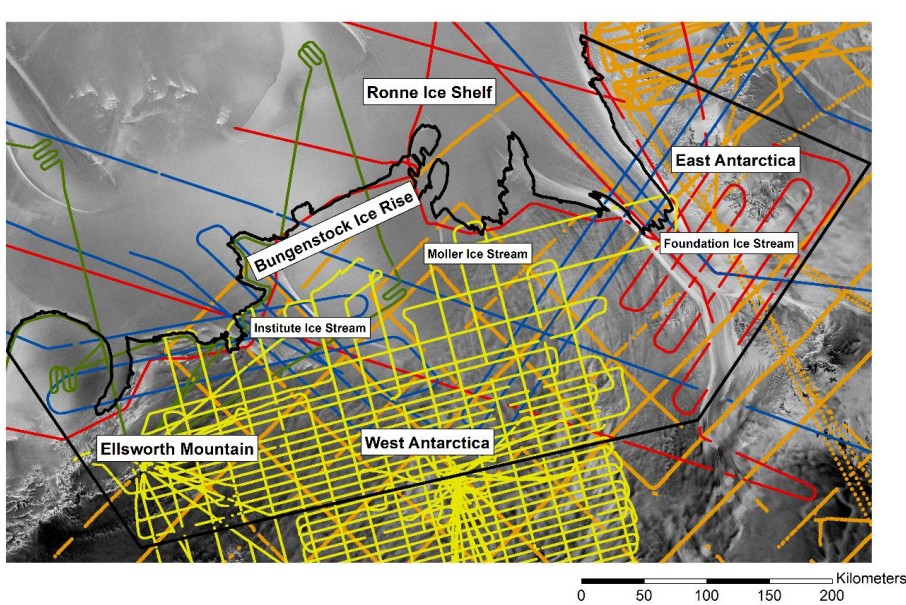



c.

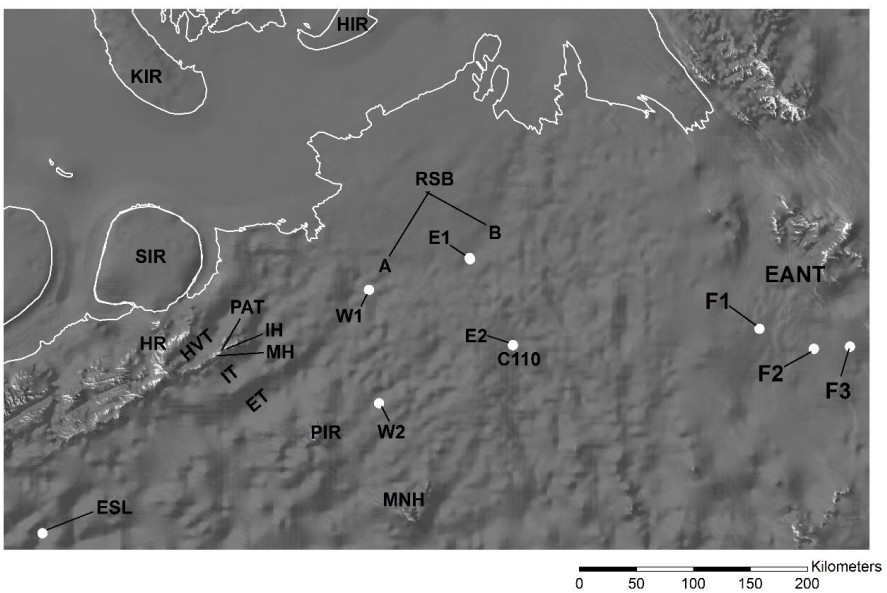

**Figure 1.** Map of: (a) A location inset of Antarctica overlain with InSAR-derived ice surface velocities (Rignot et al., 2011b); (b)The aerogeophysical flight lines across Weddell Sea sector superimposed over RADARSAT (25m) satellite imagery mosaic (Jezek, 2002); SPRI airborne survey 1969 – 1979 (orange); BAS GRADES/IMAGE survey 2008 (green); BAS Institute ice stream survey 2010/2011 (IMAFI) (yellow); OIB 2012 (red); and OIB 2014 (blue); (c) Map of the Weddell Sea sector on MODIS imagery (Haran et al., 2014); Annotations: KIR – Korff ice rise; HIR – Henry ice rise; SIR – Skytrain ice rise; HR – Heritage Range; HVT – Horseshoe Valley Trough; IT – Independence Trough; ET – Ellsworth Trough; ESL – Ellsworth Subglacial Lake; E1 – Institute E1 Subglacial Lake; E2 – Institute E2 Subglacial Lake; W1 – Institute W1 Subglacial Lake; W2 Institute W2 Subglacial Lake; F1 – Foundation 1 Subglacial Lake; F2 – Foundation 2 Subglacial Lake; F3 – Foundation 3 Subglacial Lake; PAT – Patriot Hills; IH – Independence Hills; MH – Marble Hills; PIR – Pirrit Hills; MNH – Martin-Nash Hills; RSB – Robin Subglacial Basin; and EANT – East Antarctica.



**a.**

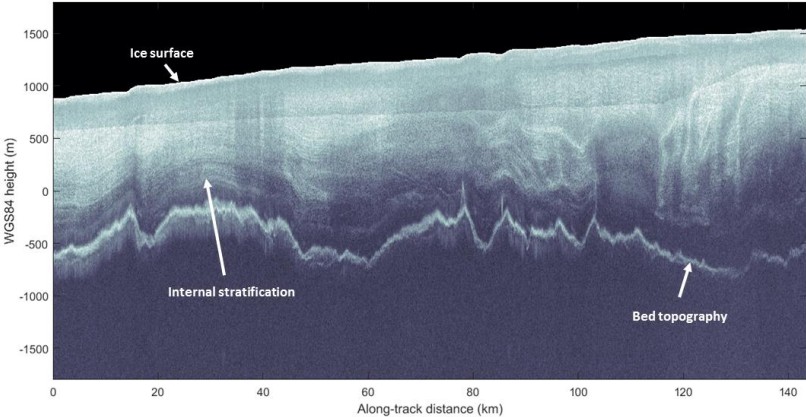

**b.**

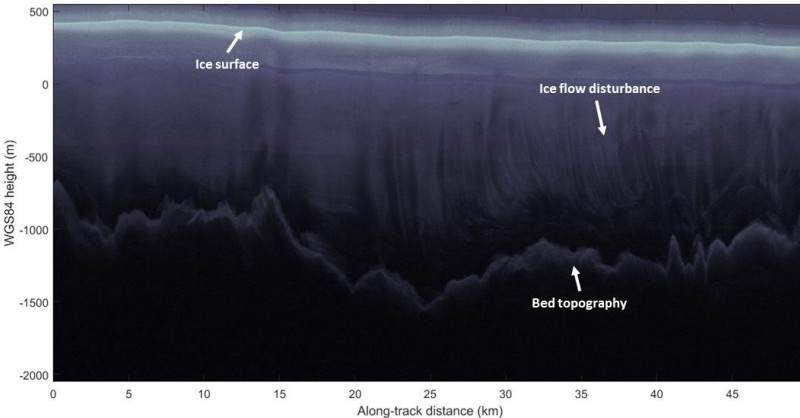

**Figure 2.** Two-dimensional synthetic aperture radar (SAR) processed radargrams of: (a) BAS IMAFI data (b) OIB data.


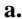

**a.**

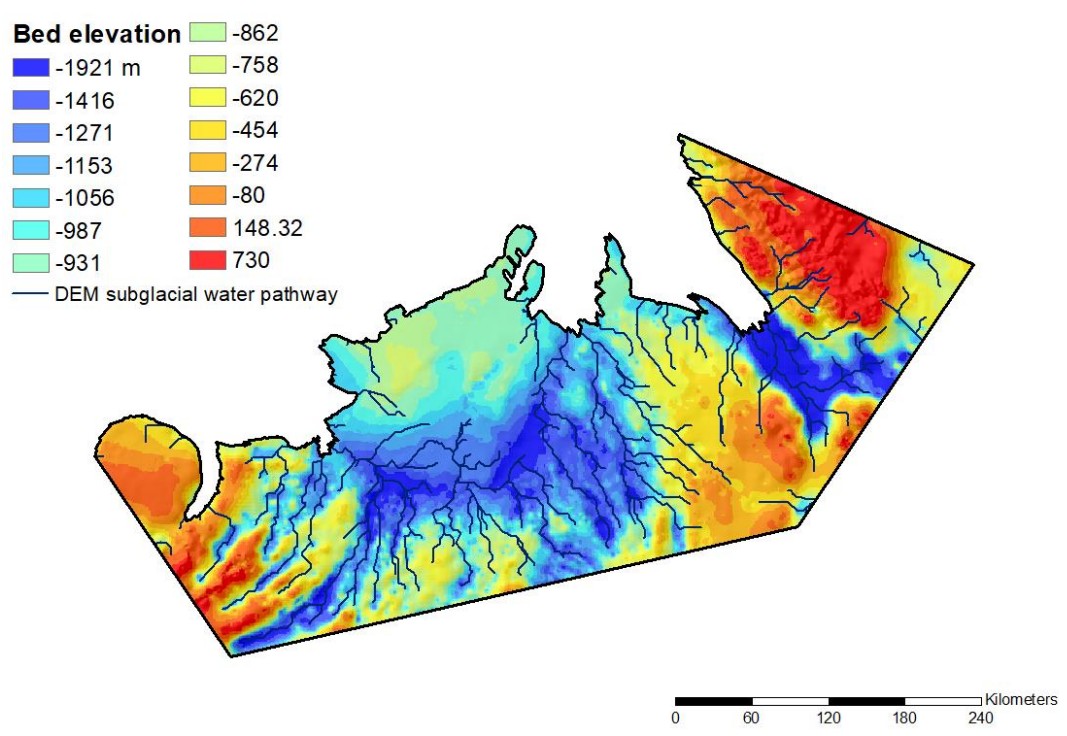

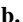

**b.**

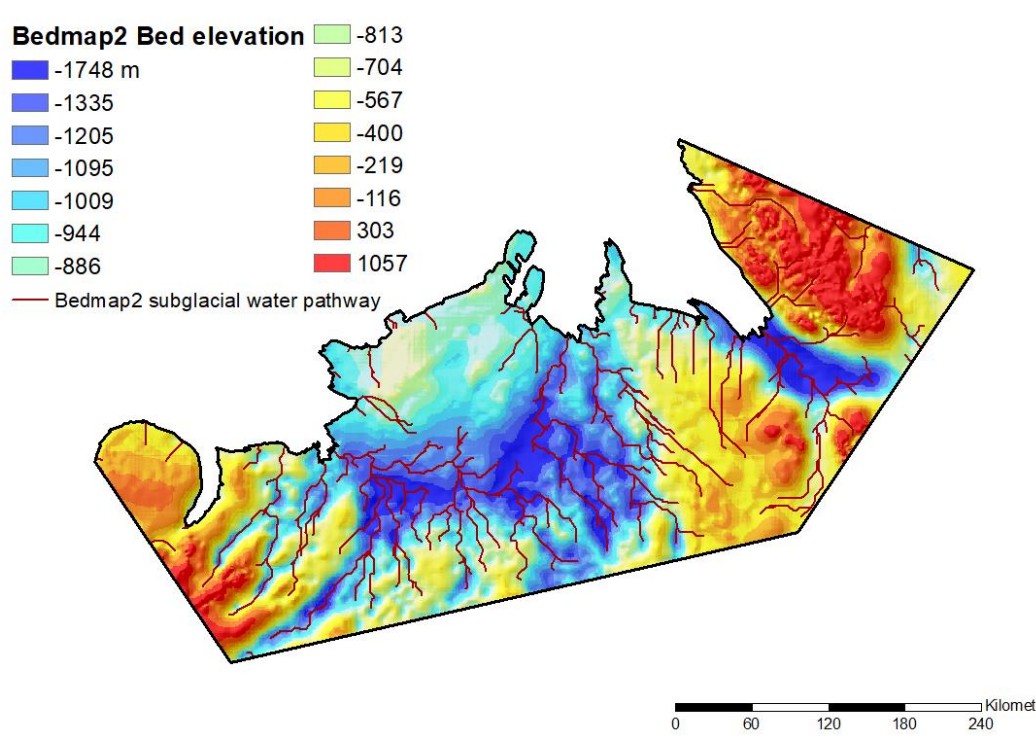





c.

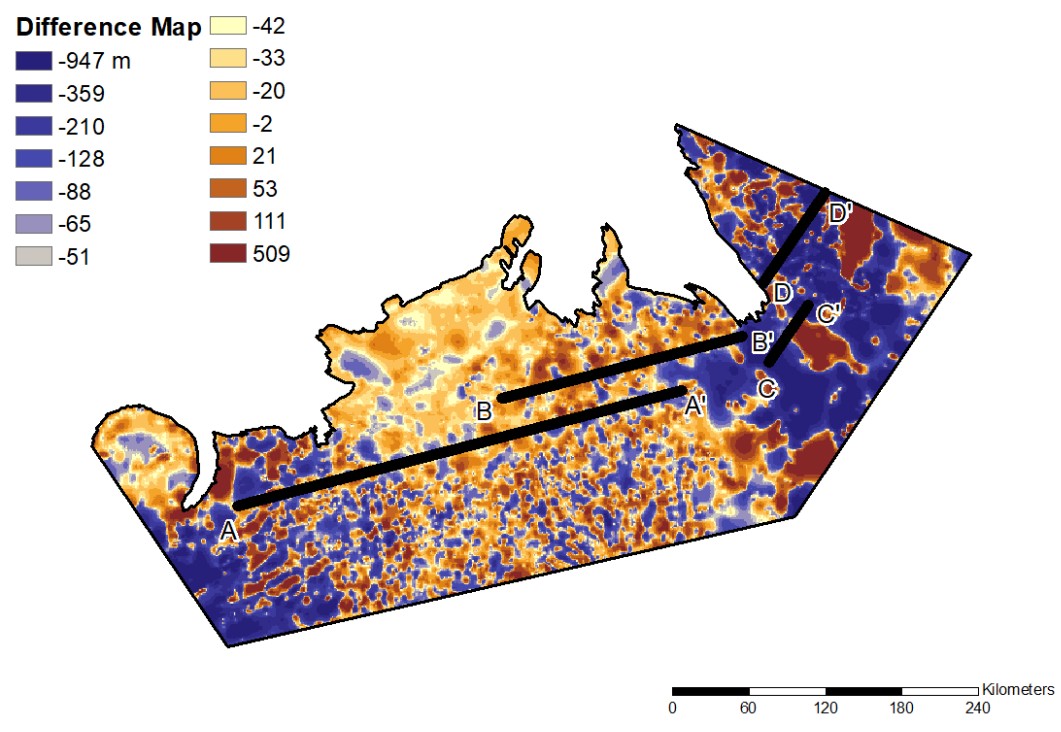

d.

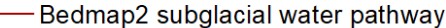

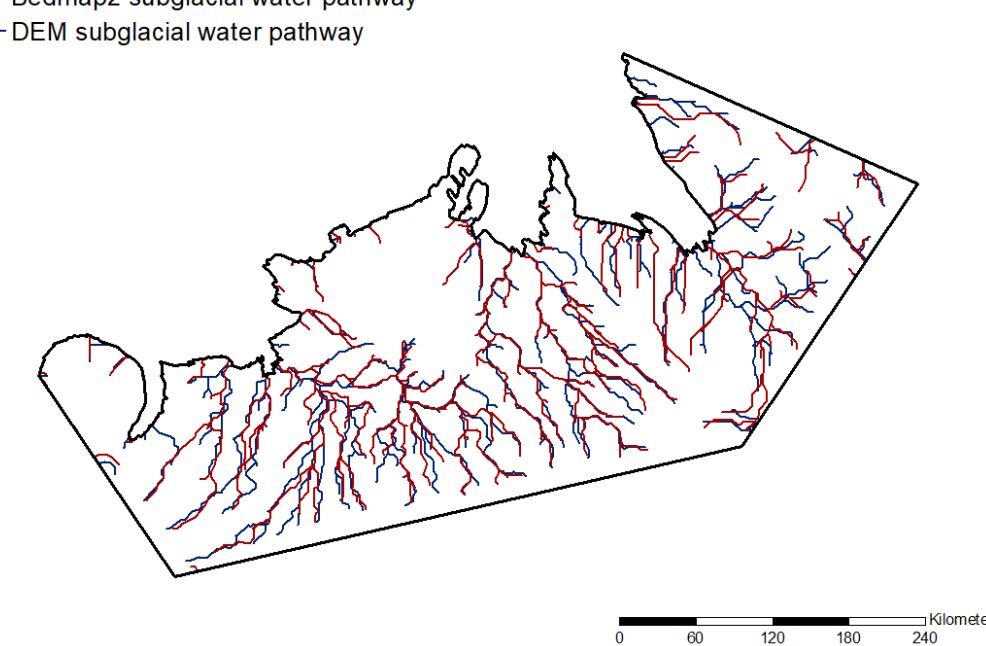



**Figure 3.** Map of: (a) A new bed topography digital elevation model; (b) Bedmap2 bed elevation product (Fretwell et al., 2013); (c) Profile A – A', B – B', C – C' and D – D' overlain by error map showing differences in bed elevation between the new DEM and Bedmap2; and (d) DEM and Bedmap2 subglacial water pathway.



**a.**

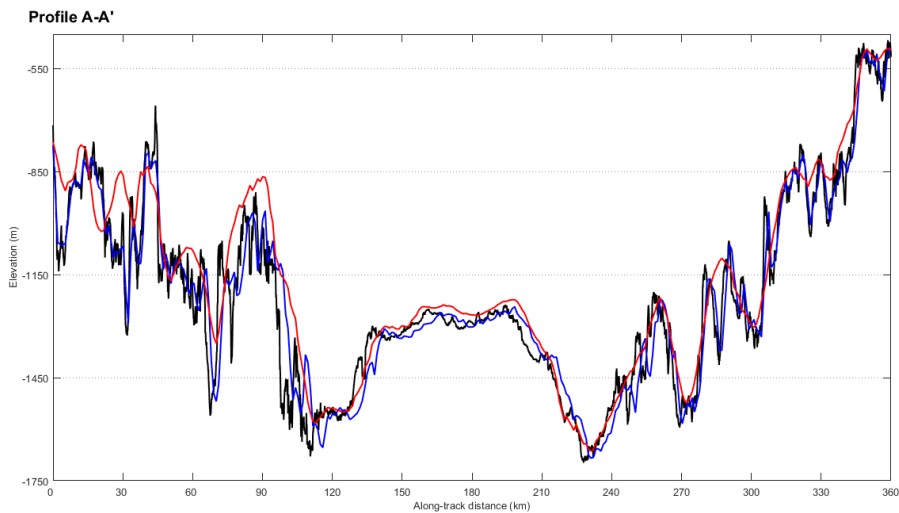

**b.**

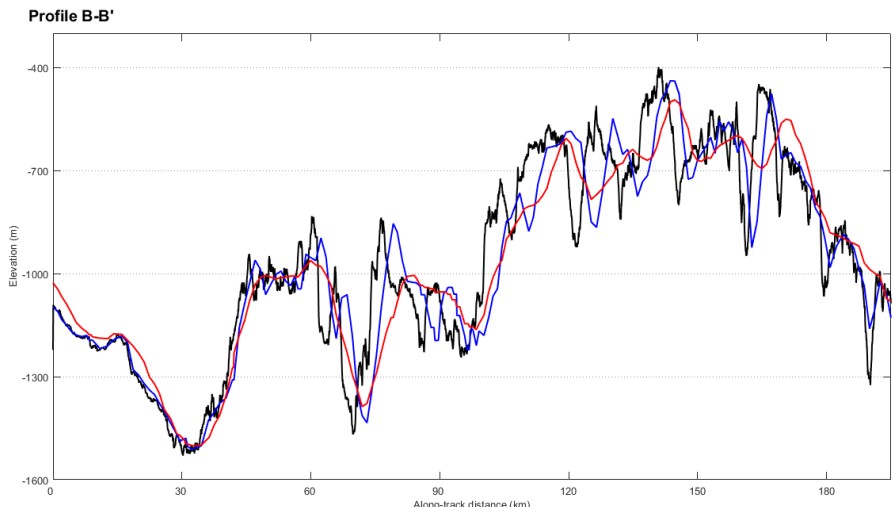



**c.**

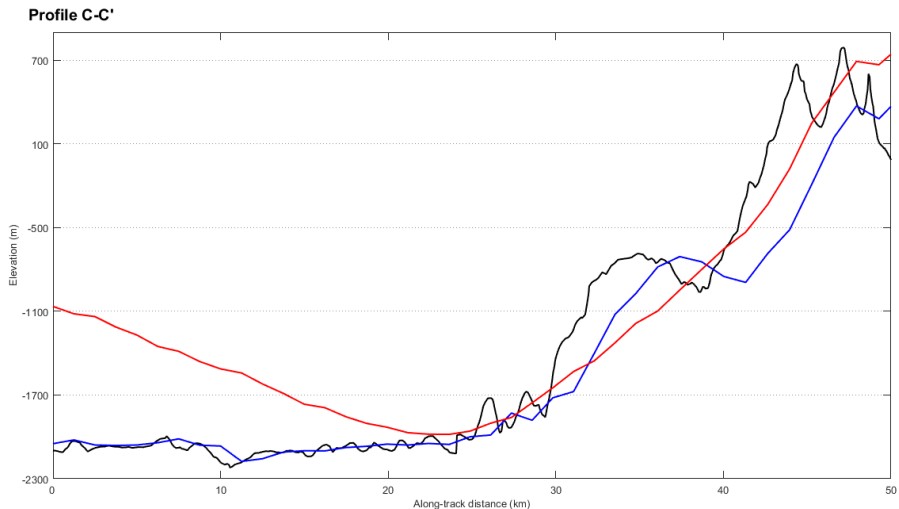

**d.**

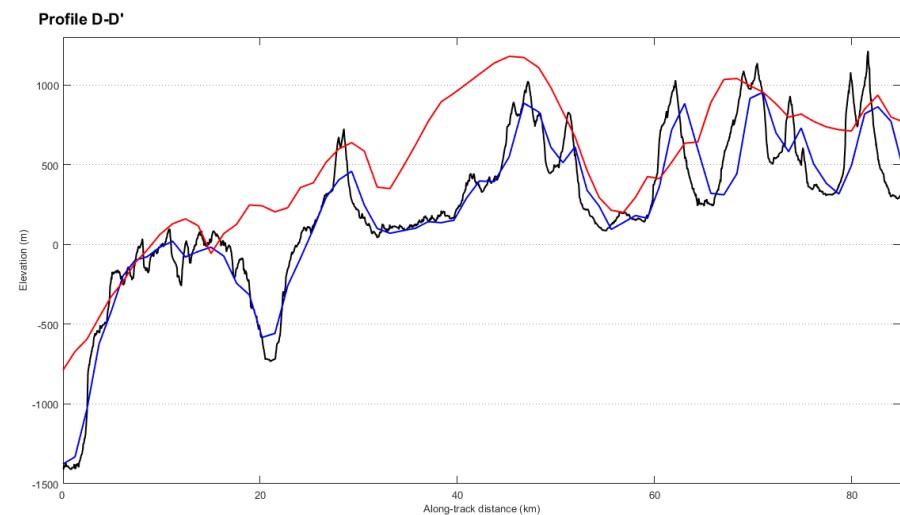

**Figure 4.** Bed elevation for the RES transect (black), DEM (blue) and Bedmap2 (red) of: (a) Profile A – A'; (b) Profile B – B'; (c) Profile C – C'; and (d) Profile D – D'.




**Table 1:** Data files and locations.

| Products | Files | Location | DOI/URL |
|---|---|---|---|
| 1-km bed elevation DEM | 1-km bed elevation DEM | Zenodo Data Repository | http://doi.org/10.5281/zenodo.1035488 |
| 1-km ice thickness DEM | Digitized SPRI-NSF-TUD bed picks data | UK Polar Data Centre (UKPDC) | https://data.bas.ac.uk/metadata.php?id= GB/NERC/BAS/AEDC/00326 |
| | BAS GRADES/IMAGE ice thickness data | UK Polar Data Centre (UKPDC) | https://secure.antarctica.ac.uk/data/aero geo/ |
| | BAS IMAFI ice thickness data | UK Polar Data Centre (UKPDC) | https://secure.antarctica.ac.uk/data/aero geo/ |
| | NASA Operation IceBridge radar depth sounder Level 2 (L2) data | Center for Remote Sensing of Ice Sheet (CReSIS) | https://data.cresis.ku.edu/ |
| 1-km ice sheet surface DEM | ERS-1 radar and ICESat laser satellite altimetry | National Snow and Ice Data Center (NSIDC) | https://nsidc.org/data/docs/daac/nsidc04 22_antarctic_1km_dem/ |
| Two-dimensional synthetic aperture radar (SAR) processed radargrams | BAS IMAFI airborne survey | UK Polar Data Centre (UKPDC) | doi.org/10.5285/8a975b9e-f18c-4c51-9bdb-b00b82da52b8 |
| | NASA Operation IceBridge airborne survey | Center for Remote Sensing of Ice Sheet (CReSIS) | https://data.cresis.ku.edu/ |
| Ice velocity map of Central Antarctica | MEaSUREs InSAR-based ice velocity | National Snow and Ice Data Center (NSIDC) | doi:10.5067/MEASURES/CRYOSPHERE /nsidc-0484.001 |
| Ice sheet surface satellite imagery | MODIS Mosaic of Antarctica (2008 – 2009) (MOA2009) | National Snow and Ice Data Center (NSIDC) | doi.org/10.7265/N5KP8037 |
| | RADARSAT (25m) satellite imagery | Byrd Polar and Climate Research Center | http://research.bpcrc.osu.edu/rsl/radarsat/d ata/ |



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
