# Peer review of "A new bed elevation model for the Weddell Sea sector of the West"

_Earth System Science Data, 2017_

## Referee Comment (RC1) · Anonymous Referee #1 · 20 Nov 2017

This is a solid contribution with a few minor issues:

Line 24: "This has" or "These have"

Lines 56-59: It's not clear what purpose this paragraph serves. Consider cutting.

Line 188: Your analysis, rather than RES, assumes this. RES can be done with other assumptions. Clarify.

Line 343: Capitalize "Institute Ice Stream"

---

## Referee Comment (RC2) · Anonymous Referee #2 · 2 Dec 2017

30 November 2017

Summary

This manuscript describes a new compilation of the bed topography of an important marine-based sector of the West Antarctic Ice Sheet. This compilation aims to supersede that of Bedmap2 by including more recent data into the same algorithm employed as that major compilation, and it considers the standard comparisons to that benchmark dataset. The physical implications of this improved bed topography are also considered, particularly in regards to subglacial hydrology and the bed elevation near the grounding line.

This is a straightforward manuscript with no significant flaws and a few minor strengths.

[Figure]

The structure is clear and consistent. The authors do an admirable job of guiding the reader through the different datasets, the subtle differences and evolutions that distinguish this study from Bedmap2, and they have a clear overall picture of the physical value of what was accomplished. The value of such a dataset rests on the coherent integration of relevant raw data, an appropriate error budget and a complete description of the resulting dataset. This manuscript fully succeeds at the first of these aspects, but falls a bit short on the latter two. My comments are mostly aimed to improving the latter two aspects.

Comments

187: An instrument cannot make assumptions, only its operators and those who interpret its data can. Further, the velocity used is an uncertainty of the order of ∼1%, because it depends on the well constrained but imperfectly known real part of the relative permittivity of pure ice and spatially variable densification. The mention of this velocity raises two important questions: 1. How is its (unstated) uncertainty incorporated into the error budget for the new bed topography? 2. Was this velocity used to correct the traveltimes between the surface and bed reflections for all datasets in this study? That's never explicitly stated. It certainly should be, and if different studies used different velocity values, then it's up to the present authors to make that important and necessary correction.

210: I am a long-time MATLAB user and I do not know what is meant by MATLAB "standard format".

218: Again, what is the origin of this estimate? Further, an error map ought to be generated and shown for the bed topography.

243-4: Explicitly state here that this is the same algorithm employed by Bedmap2, because it is somewhat primitive compared to that typically employed in the slow-moving sectors of Greenland, i.e.., ordinary kriging.

252-3: Please reconsider significant figures here.

277: Where is the Pagano Shear Zone? It's not identified in Figure 1c.

292: The significance of a change in linear correlation coefficient can be easily calculated, and it ought to be done so if the term "significantly reduced" is to be used.

297-8: This statement contradicts the use of an outdated subaerial ice-surface DEM, i.e., the same as that of the Bedmap2. However, given the figures presented, the time to accomplish this task should not be significant. A newer Cryosat-2 DEM from Helm et al. (2014, The Cryosphere) ought to be employed, as it clearly demonstrates greater fidelity to high-resolution airborne altimetry than the ICESat/ERS-1/2 DEM that Bedmap2 employed (Table 2 of that study).

315-322: The geographic coordinates of the lakes do not need to be mentioned, and if a graticule were added to Figure 1c they would become even more unnecessary. Further, this paragraph doesn't really add much information about the lakes that is not available from existing inventories. It simply enumerates them. Reconsider.

332-333: This statement about Bungenstock Ice Rise being a good example of one is not very meaningful.

336-337: This statement is wrong. There are very clear metrics for measuring surface roughness, independent of FFTs. Shepard et al. (2001, JGR) summarizes them very nicely, and they have been employed in several glaciological studies of ice-sheet beds (e.g., Young et al., 2011, Nature ; MacGregor et al., 2013, JGlac).

Figures & Tables

All the figures in this study need significant improvements.

Figure 1. (a) Add a scale bar, the grounding line and a color bar for the surface velocity. (b) Add a legend for all the different surveys. (c). This panel is quite hard to read. The black labels on a dark gray background do not work well. Brighten MOA and increase

the contrast. It is mentioned in the text that different grounding lines exist, and all should be shown on this figure. Also, a distinct symbology should be used for different types of features (e.g., lakes vs. troughs, rather than using the same white dot for everything.

Figure 2. It is very difficult to make out much of anything on the lower radargram. Adjust its color scale.

Figure 3. (a,b) I recommend using the USGS color scale for topography instead. demcmap in MATLAB. The bizarre irregular intervals for the color scale are unacceptable. Use simple, regular intervals, e.g., –2000:200:600 in MATLAB form. (c) Again, fix the weird irregular intervals. I'm not sure why the red/blue color scale with yellow was invented, but it should be replaced with one that has red/blue with white in the middle. Much more intuitive.

Figure 4. Add a legend. There's plenty of room for one. These figures aren't information-dense, so generate them closer together.

Grammar, etc.

121: C-130R or LC-130? 143: I do not understand why "chirp" is capitalized here 165: to be 7 cm 200: radar shot number that is used 210: MATLAB should be capitalized 279: Here and throughout the manuscript, capitalize all proper noun geographic locations, e.g., Institute Ice Stream 345: Passive voice: "It is considered. . ."

---

## Author Comment (AC1) · 30 Jan 2018

Dear Sir/Madam,

Many thanks for reviewing the above paper. The comments are positive, reasonable and helpful. We have revised and made some adjustments to the paper accordingly, and below we detail the changes that have been made.

We hope that the paper is now ready for publication in Earth System Science Data.

Sincerely,

Hafeez Jeofry (and on behalf of co-authors)

Referee #1

[Figure]

Line 24: "This has" or "These have"

Noted, however, the statement has been removed.

Lines 56-59: It's not clear what purpose this paragraph serves. Consider cutting.

Noted, the paragraph has been deleted.

Line 188: Your analysis, rather than RES, assumes this. RES can be done with other assumptions. Clarify.

Noted, the statement has been change to "We assume Airborne RES propagates..."

Line 343: Capitalize "Institute Ice Stream"

Noted, change has been made.

Referee #2

Line 187: An instrument cannot make assumptions, only its operators and those who interpret its data can. Further, the velocity used is an uncertainty of the order of 1%, because it depends on the well constrained but imperfectly known real part of the relative permittivity of pure ice and spatially variable densification. The mention of this velocity raises two important questions: 1. How is its (unstated) uncertainty incorporated into the error budget for the new bed topography? 2. Was this velocity used to correct the travel times between the surface and bed reflections for all datasets in this study? That's never explicitly stated. It certainly should be, and if different studies used different velocity values, then it's up to the present authors to make that important and necessary correction.

Noted, the statement has been change to "We assume Airborne RES propagates..."

All of the CReSIS, BAS GRADES/IMAGE and IMAFI surveys used a constant radio-wave speed of 0.168 m/ns. CReSIS OIB used 3.15 as the ice dielectric permittivity without considering the effect of firn on the top, so the radio wave speed in ice is c/sqrt (3.15), where c is the light speed in air. The thickness error using a constant ice dielectric permittivity is usually within + 10 m at deep depth. Discussion about the thickness error for CReSIS OIB radar is discussed in "Coherent radar ice thickness measurements over the Greenland ice sheet (Gogineni et al.)". As for the BAS surveys, a nominal value of 10 m is used to correct for the firn layer during the processing of ice thickness and this introduces an error of the order of +3 m.

In summary, we have made this issue clear in the text.

Line 210: I am a long-time MATLAB user and I do not know what is meant by MATLAB "standard format".

Noted, we have change the "standard format" to "data binary files".

Line 218: Again, what is the origin of this estimate? Further, an error map ought to be generated and shown for the bed topography.

The error was derived from the crossover analysis which showed that the RMS differences is 18.29 m ice thickness. The quoted RMS error should roughly approximate +1% of total ice thickness. The purpose for the estimate is simply to acknowledge the error due to the firn layer and, so, we do not feel an error map needs to be generated.

Line 243-4: Explicitly state here that this is the same algorithm employed by Bedmap2, because it is somewhat primitive compared to that typically employed in the slow-moving sectors of Greenland, i.e., ordinary kriging.

Noted, changes have been made.

Line 252-3: Please reconsider significant figures here.

Noted, changes have been made.

Line 277: Where is the Pagano Shear Zone? It's not identified in Figure 1c.

Noted, we have removed "Pagano Shear Zone" from the text.
Line 292: The significance of a change in linear correlation coefficient can be easily calculated, and it ought to be done so if the term "significantly reduced" is to be used.

Noted, we have calculated the relative errors between the correlation coefficients of our new DEM and Bedmap2 product for each Profile. The relative errors are 2%, 1%, 12% and 13% for Profile A, B, C and D, respectively. The changes have been added to the text as well which is as follow:

"These value contrast with correlation coefficients from Bedmap2 of 0.87 for Profile C and 0.83 for Profile D, with relative errors of 12 and 13%, respectively."

Line 297-8: This statement contradicts the use of an outdated subaerial ice-surface DEM, i.e., the same as that of the Bedmap2. However, given the figures presented, the time to accomplish this task should not be significant. A newer Cryosat-2 DEM from Helm et al. (2014, The Cryosphere) ought to be employed, as it clearly demonstrates greater fidelity to high-resolution airborne altimetry than the ICESat/ERS-1/2 DEM that Bedmap2 employed (Table 2 of that study).

Point well made, however, no change has been made. There are two reasons why we did not consider the newer ice-surface Cryosat-2 DEM to compute the subglacial water pathway:

1. It is true that hydrology pathway is highly sensitive to ice-surface elevation relative to bed topography. However, by using the same ice-surface DEM to generate our new bed DEM as the ice-surface DEM used in Bedmap2, we are able to compute hydrology pathways and observe how the water routing has changed as a consequence of the new bed measurements (rather than ice surface change). In doing to, we are able to pinpoint several local small-scale differences in the water pathways between the respective bed topographies, which highlights hydraulic sensitivity in the region.

2. While the main product of this paper is the 1-km bed elevation DEM of the Weddell Sea sector, we also include 1-km ice thickness DEM of the Weddell Sea sector into the data repository, which allows others to modify or improve the DEM as new ice surface information appears. The bed elevation derived using mass conservation and kriging, as shown in figure 3e, is an example of how we can further improve the DEM.

Line 315-322: The geographic coordinates of the lakes do not need to be mentioned, and if a graticule were added to Figure 1c they would become even more unnecessary. Further, this paragraph doesn't really add much information about the lakes that is not available from existing inventories. It simply enumerates them. Reconsider.

Noted, we have deleted the geographic coordinates and include the graticule to Figure 1c.

Line 332-333: This statement about Bungenstock Ice Rise being a good example of one is not very meaningful.

Noted, we have deleted the statement.

Line 336-337: This statement is wrong. There are very clear metrics for measuring surface roughness, independent of FFTs. Shepard et al. (2001, JGR) summarizes them very nicely, and they have been employed in several glaciological studies of ice-sheet beds (e.g., Young et al., 2011, Nature; MacGregor et al., 2013, JGlac).

Point well made, we have deleted "Although there is no specific method or standardized unit to measure bed roughness,".

Figures & Tables

All the figures in this study need significant improvements. Figure 1. (a) Add a scale bar, the grounding line and a colour bar for the surface velocity. (b) Add a legend for all the different surveys. (c). This panel is quite hard to read. The black labels on a dark gray background do not work well. Brighten MOA and increase the contrast. It is mentioned in the text that different grounding lines exist, and all should be shown on this figure. Also, a distinct symbology should be used for different types of features (e.g., lakes vs. troughs, rather than using the same white dot for everything.

Figure 1a – We have added the scale bar, the MOA grounding line and the colour bar for the surface velocity.

Figure 1b – We have added the legend.

Figure 1c – We have added a white background to the labels, and we have added all four grounding lines. Note that we did not put legends of the different grounding lines due to limited space on the figure. However, we have explained and distinguished the different grounding lines based on its colour in the caption. Finally, we used a different symbology for lakes (white dot) and troughs (white asterisk). A graticule has also been added.

Figure 2. It is very difficult to make out much of anything on the lower radargram. Adjust its colour scale.

Noted, we have adjusted the colour scale of the lower radargram.

Figure 3. (a,b) – I recommend using the USGS color scale for topography instead. Demcmap in MATLAB. The bizarre irregular intervals for the colour scale are unacceptable. Use simple, regular intervals, e.g., –2000:200:600 in MATLAB form. (c) Again, fix the weird irregular intervals. I'm not sure why the red/blue color scale with yellow was invented, but it should be replaced with one that has red/blue with white in the middle. Much more intuitive.

We have adjusted the interval for figure 3a and 3b using -2000:200:600 and for figure 3c using -350:50:350 intervals. However, no change has been made to the colour scale itself. We have tried the suggested colour scale, but it is not appealing to us and we have no problem with the current colour scale.

Figure 4. Add a legend. There's plenty of room for one. These figures aren't information-dense, so generate them closer together.

We have added the legends in each profile figure and have arranged them close together.

Grammar

Line 121: C-130R or LC-130?

Noted, it is LC-130, change has been made.

Line 143: I do not understand why "chirp" is capitalized here

Noted, change has been made.

Line 165: to be 7 cm

Noted, change has been made.

Line 200: radar shot number that is used

Noted, change has been made.

Line 210: MATLAB should be capitalized

Noted, change has been made.

Line 279: Here and throughout the manuscript, capitalize all proper noun geographic locations, e.g., Institute Ice Stream

Noted, change has been made.

Line 345: Passive voice: "It is considered. . ."

Noted, however, no change has been made.

**Further editing**

Major Issues

1.

We have added a new method in section 4.4 "Inferring bed elevation using mass conservation (MC) and Kriging" and a new figure 3e which is the modified version of our new bed elevation DEM product using MC at the fast-flowing region of ice and Kriging at the slow-moving ice:

"Relying on the conservation of mass (MC) to infer the bed between flight lines (Morlighem et al., 2011), we were able to investigate how the bed can be developed further in fast-flowing regions, using a new interpolation technique. To perform the MC procedure, we used InSAR-derived surface velocities from Rignot et al. (2011b), surface mass balance from RACMO 2.3 (Van Wessem et al., 2014), and assumed that the ice thinning/thickening rate and basal melt are negligible. We constrained the optimization with ground penetrating radar from CReSIS, GRADES, IMAFI and SPRI, described above, and used a mesh horizontal resolution of 500 m."

The purpose of this added product is to demonstrate that there are many ways that our new DEM can be used and modified according to their interpretations of the ice-sheet dynamic in the WS sector region. We have added a new paragraph explaining the details of the new product in section 5.1:

"The new DEM can be refined further to deal with bumps and irregularities associated with interpolation effects from along-track data in otherwise data sparse regions. We used mass conservation (MC) and kriging to infer the bed elevation (Fig. 3e) beneath the fast and slow-moving ice, respectively, similar to that successfully employed in Greenland (Morlighem et al., 2017). In general, the bed elevation derived from MC and kriging is consistent with our new DEM. However, there are significant changes in the bed morphology beneath fast flowing ice. For example, the tributary of the Foundation Ice Stream has been extended for $\sim$100 km further inland relative to our new DEM."

2.

We have made corrections to account for the way Bedmap2 is projected using the gl04c geoid. (Bedmap2 was produced relative to the gl04c geoid to allow realistic numerical modelling by incorporating the role of relative sea level / glacial isostatic adjustment.) In a previous figure, we did not correct the datum and, so, it was not strictly comparable with our new DEM, because this is produced relative to the WGS 1984 Antarctic Polar Stereographic projection. We have made the necessary corrections by transforming Bedmap2 onto the WGS 1984 Antarctic Polar Stereographic. Consequently, we have reproduced the figure 3c "Difference map" and figure 4. The re-analysis does not have a major difference relative to the previous figures, however. Finally, we have made the necessary adjustments to the figures, and have added text into the methodology section 4.3 as follows:

"The Bedmap2 bed elevation product (Fretwell et al., 2013) was transformed from the gl04c geoid projection to the WGS 1984 Antarctic Polar Stereographic projection."

Minor Issues

Line 21: "update" to "improve"

Line 23-24: "as BEDMAP2 included only relatively crude ice thickness measurements determined in the field for quality control" to "from the relatively crude measurements determined in the field for quality control purposes used in Bedmap2".

Line 24-25: "This have resulted in the deep parts of the topography not being visible in the fieldwork non-SAR processed radargrams" deleted.

Line 26: "deep trough" to "deep subglacial trough".

Line 28: "at the ice-bed interface" deleted.

Line 29: "sensitivity" to "potential vulnerability".

Line 29: "of the ice sheet" deleted.

Line 32: "in the" to "at".

Line 37: "projected" to "concluded".

Line 41: "in places more than" added.

Line 43: "future" deleted.

Line 44: "project" to "show".

Line 46: "similar manner" to "manner similar".

Line 48-49: "from the parent ice mass" deleted.

Line 49: "together" deleted.

Line 62: "entire" deleted.

Line 64: "late 1960s and.." added.

Line 66: "and" deleted.

Line 66: "thickness" to "ice thickness"

Line 68: "highly" deleted.

Line 74: "line representation" to "lines compared".

Line 74: "observations" to "satellite-derived observations".

Line 75: "limited" to "restricted".

Line 75: "to the community" deleted.

Line 77: "project" to "compilation".

Line 79: "where only 34% of cells have data and 80% have data within the range of 20 km" deleted.

Line 85: "improving knowledge of the land surface beneath the ice sheet. This. . ." to ". . .,which. . ."

Line 87: "the" deleted.

Line 87: "of" to "associated with".

Line 88: "in glaciology" added.

[Figure]

Line 90: "a revised" to "an improved".

Line 96: "The study area is located in" deleted.

Line 96: "covering" to "covers".

Line 97: "The DEM" to "The region covered by the DEM".

Line 98: "with the" to "with".

Line 102: "We used the Differential Interferometry Synthetic Aperture Radar (DInSAR) grounding line (Rignot et al., 2011b) to delimit the ice shelf-facing margin of our grid" added.

Line 102-107: "The data and methods used in this study are mostly similar to Jeofry et al. (2017), however, we have expanded the boundaries of our new bed elevation DEM by focussing across the Weddell Sea sector relative to the Institute ice stream as the main focus in Jeofry et al. (2017). In addition, this paper focuses on discussing the geomorphology and improvement that have been made between our new bed elevation DEM and Bedmap2 product whereas Jeofry et al. (2017) discuss on the hitherto unknown subglacial embayment and its effect to the ice sheet dynamics" deleted.

Line 120: "of the 1970s" deleted.

Line 132: "was" to "were".

Line 133: "was" to "were".

Line 134: "records were" to "record was".

Line 144: "is" to "are".

Line 157: the coordinate for C110 "-82o 37' 30" N, 280o 59' 13.2"E" has been removed. It has been replaced with "17 flights were flown from C110, which is located above Institute E2 subglacial lake,. . .".

Line 158: the coordinate for Patriot Hills "-80o 19' 60" N, 278o 34' 60"E" has been removed.

Line 199: "200m" to "200 m".

Line 247-249: "The ice thickness DEM was then subtracted from the ice-sheet surface elevation derived from the combined European Remote Sensing Satellite-1 (ERS-1) radar and Ice, Cloud and land Elevation Satellite (ICESat) laser satellite altimetry DEM (Bamber et al., 2009a), to derive the bed topography" to "The ice thickness DEM was then subtracted from the ice-sheet surface elevation DEM (from European Remote Sensing Satellite-1 (ERS-1) radar and Ice, Cloud and land Elevation Satellite (ICESat) laser satellite altimetry datasets (Bamber et al., 2009a)), to derive the bed topography".

Line 289: "the" deleted.

Line 291: "the" deleted.

Line 298: "It is noted that" deleted.

Line 304: "this is" deleted.

Line 310: "almost" deleted.

Line 312: "a vital constituent" to "an obvious component".

Line 313: "melts" to "allows"

Line 314: "to melt if it is >∼2km" added.

Line 314: "on the bed" added.

Line 314: "of the ice" deleted.

Line 315: "At the western margin of the Ellsworth Mountain" to "West of the Ellsworth Mountain".

Line 315: "large" deleted.

Line 317: "In some cases, ice-surface elevation changes have been linked with subglacial lake hydrological change (Siegert et al., 2016a) – the so called 'active' subglacial lakes." Added.

Line 317: "active" added.

Line 318: "The" deleted

Line 319: "The" deleted.

Line 321: "active" added.

Line 323: "East of the Foundation Ice Stream, there are 16 active subglacial lakes distributed along the main trunk of Academy Ice Stream (Wright and Siegert, 2012)" added.

Line 340: "of the" deleted.

Line 343: "and its connection" to "and connects".

Line 350: "It is worth noting that the lake is outside the grid of our new DEM" added.

Line 385: "in the" to "at".

Line 416: "the Institute ice stream, Bungenstock ice rise, Möller and Foundation ice streams" to "the Institute, Möller and Foundation ice streams, and the Bungenstock ice rise".

Line 417: "between the new DEM and that of the previous DEM of the region (Bedmap2)" to "between the new and previous DEMs (i.e., Bedmap2)"

Line 419-421: "the bed elevation of our DEM appears to be slightly lower relative to the bed elevation of Bedmap2 DEM, which is likely the results of the deep sections of the topography not being visible in the fieldwork non-SAR processed radargrams" to "improved processing of existing data has led to some particularly deep regions of bed to be better resolved than in Bedmap2"

Line 422: "that derived from" deleted.

Please also note the supplement to this comment:
https://www.earth-syst-sci-data-discuss.net/essd-2017-90/essd-2017-90-AC1-
supplement.pdf

**Supplement:**

[Figure]

**Grantham Institute and Department of Earth Science and Engineering**
Imperial College London

South Kensington Campus
London SW7 2AZ

**Muhammad Hafeez Jeofry**
PhD student h.jeofry@imperial.ac.uk

January 2018

**Response to Reviewers' comments on manuscript #essd-2017-90**

Dear Sir/Madam,

Many thanks for reviewing the above paper. The comments are positive, reasonable and helpful. We have revised and made some adjustments to the paper accordingly, and below we detail the changes that have been made.

We hope that the paper is now ready for publication in Earth System Science Data.

Sincerely,

Hafeez Jeofry (and on behalf of co-authors)
* * *
**Referee #1**

**Line 24: "This has" or "These have"**

Noted, however, the statement has been removed.

**Lines 56-59: It's not clear what purpose this paragraph serves. Consider cutting.**

Noted, the paragraph has been deleted.

**Line 188: Your analysis, rather than RES, assumes this. RES can be done with other assumptions. Clarify.**

Noted, the statement has been change to "*We assume Airborne RES propagates…*"

**Line 343: Capitalize "Institute Ice Stream"**

Noted, change has been made.
* * *
**Referee #2**

**Line 187: An instrument cannot make assumptions, only its operators and those who interpret its data can. Further, the velocity used is an uncertainty of the order of 1%, because it depends on the well constrained but imperfectly known real part of the relative permittivity of pure ice and spatially variable densification. The mention of this velocity raises two important questions: 1. How is its (unstated) uncertainty incorporated into the error budget for the new bed topography? 2. Was this velocity used to correct the travel times between the surface and bed reflections for all datasets in this study? That's never explicitly stated. It certainly should be, and if different studies used different velocity values, then it's up to the present authors to make that important and necessary correction.**

Noted, the statement has been change to "*We assume Airborne RES propagates…*"

All of the CReSIS, BAS GRADES/IMAGE and IMAFI surveys used a constant radio-wave speed of 0.168 m/ns. CReSIS OIB used 3.15 as the ice dielectric permittivity without considering the effect of firn on the top, so the radio wave speed in ice is c/sqrt (3.15), where c is the light speed in air. The thickness error using a constant ice dielectric permittivity is usually within $\pm$ 10 m at deep depth. Discussion about the thickness error for CReSIS OIB radar is discussed in "Coherent radar ice thickness measurements over the Greenland ice sheet (Gogineni et al.)". As for the BAS surveys, a nominal value of 10 m is used to correct for the firn layer during the processing of ice thickness and this introduces an error of the order of $\pm$3 m.

In summary, we have made this issue clear in the text.

**Line 210: I am a long-time MATLAB user and I do not know what is meant by MATLAB "standard format".**

Noted, we have change the "standard format" to "data binary files".

**Line 218: Again, what is the origin of this estimate? Further, an error map ought to be generated and shown for the bed topography.**

The error was derived from the crossover analysis which showed that the RMS differences is 18.29 m ice thickness. The quoted RMS error should roughly approximate $\pm$1% of total ice thickness. The purpose for the estimate is simply to acknowledge the error due to the firn layer and, so, we do not feel an error map needs to be generated.

**Line 243-4: Explicitly state here that this is the same algorithm employed by Bedmap2, because it is somewhat primitive compared to that typically employed in the slow-moving sectors of Greenland, i.e., ordinary kriging.**

Noted, changes have been made.

**Line 252-3: Please reconsider significant figures here.**

Noted, changes have been made.

**Line 277: Where is the Pagano Shear Zone? It's not identified in Figure 1c.**

Noted, we have removed "Pagano Shear Zone" from the text.

**Line 292: The significance of a change in linear correlation coefficient can be easily calculated, and it ought to be done so if the term "significantly reduced" is to be used.**

Noted, we have calculated the relative errors between the correlation coefficients of our new DEM and Bedmap2 product for each Profile. The relative errors are 2%, 1%, 12% and 13% for Profile A, B, C and D, respectively. The changes have been added to the text as well which is as follow:

*"These value contrast with correlation coefficients from Bedmap2 of 0.87 for Profile C and 0.83 for Profile D, with relative errors of 12 and 13%, respectively."*

**Line 297-8: This statement contradicts the use of an outdated subaerial ice-surface DEM, i.e., the same as that of the Bedmap2. However, given the figures presented, the time to accomplish this task should not be significant. A newer Cryosat-2 DEM from Helm et al. (2014, The Cryosphere) ought to be employed, as it clearly demonstrates greater fidelity to high-resolution airborne altimetry than the ICESat/ERS-1/2 DEM that Bedmap2 employed (Table 2 of that study).**

Point well made, however, no change has been made. There are two reasons why we did not consider the newer ice-surface Cryosat-2 DEM to compute the subglacial water pathway:

1. It is true that hydrology pathway is highly sensitive to ice-surface elevation relative to bed topography. However, by using the same ice-surface DEM to generate our new bed DEM as the ice-surface DEM used in Bedmap2, we are able to compute hydrology pathways and observe how the water routing has changed as a consequence of the new bed measurements (rather than ice surface change). In doing to, we are able to pinpoint several local small-scale differences in the water pathways between the respective bed topographies, which highlights hydraulic sensitivity in the region.
2. While the main product of this paper is the 1-km bed elevation DEM of the Weddell Sea sector, we also include 1-km ice thickness DEM of the Weddell Sea sector into the data repository, which allows others to modify or improve the DEM as new ice surface information appears. The bed elevation derived using mass conservation and kriging, as shown in figure 3e, is an example of how we can further improve the DEM.

**Line 315-322: The geographic coordinates of the lakes do not need to be mentioned, and if a graticule were added to Figure 1c they would become even more unnecessary. Further, this paragraph doesn't really add much information about the lakes that is not available from existing inventories. It simply enumerates them. Reconsider.**

Noted, we have deleted the geographic coordinates and include the graticule to Figure 1c.

**Line 332-333: This statement about Bungenstock Ice Rise being a good example of one is not very meaningful.**

Noted, we have deleted the statement.

**Line 336-337: This statement is wrong. There are very clear metrics for measuring surface roughness, independent of FFTs. Shepard et al. (2001, JGR) summarizes them very nicely, and they have been employed in several glaciological studies of ice-sheet beds (e.g., Young et al., 2011, Nature; MacGregor et al., 2013, JGlac).**

Point well made, we have deleted "*Although there is no specific method or standardized unit to measure bed roughness,*".
* * *
**Figures & Tables**

**All the figures in this study need significant improvements.**
**Figure 1. (a) Add a scale bar, the grounding line and a colour bar for the surface velocity. (b) Add a legend for all the different surveys. (c). This panel is quite hard to read. The black labels on a dark gray background do not work well. Brighten MOA and increase the contrast. It is mentioned in the text that different grounding lines exist, and all should be shown on this figure. Also, a distinct symbology should be used for different types of features (e.g., lakes vs. troughs, rather than using the same white dot for everything.**

Figure 1a – We have added the scale bar, the MOA grounding line and the colour bar for the surface velocity.
Figure 1b – We have added the legend.
Figure 1c – We have added a white background to the labels, and we have added all four grounding lines. Note that we did not put legends of the different grounding lines due to limited space on the figure. However, we have explained and distinguished the different grounding lines based on its colour in the caption. Finally, we used a different symbology for lakes (white dot) and troughs (white asterisk). A graticule has also been added.

**Figure 2. It is very difficult to make out much of anything on the lower radargram. Adjust its colour scale.**

Noted, we have adjusted the colour scale of the lower radargram.

**Figure 3. (a,b) I recommend using the USGS color scale for topography instead. Demcmap in MATLAB. The bizarre irregular intervals for the colour scale are unacceptable. Use simple, regular intervals, e.g., –2000:200:600 in MATLAB form. (c) Again, fix the weird irregular intervals. I'm not sure why the red/blue color scale with yellow was invented, but it should be replaced with one that has red/blue with white in the middle. Much more intuitive.**

We have adjusted the interval for figure 3a and 3b using -2000:200:600 and for figure 3c using -350:50:350 intervals. However, no change has been made to the colour scale itself. We have tried the suggested colour scale, but it is not appealing to us and we have no problem with the current colour scale.

**Figure 4. Add a legend. There's plenty of room for one. These figures aren't information-dense, so generate them closer together.**

We have added the legends in each profile figure and have arranged them close together.

**Grammar**

**Line 121: C-130R or LC-130?**

Noted, it is LC-130, change has been made.

**Line 143: I do not understand why "chirp" is capitalized here**

Noted, change has been made.

**Line 165: to be 7 cm**

Noted, change has been made.

**Line 200: radar shot number that is used**

Noted, change has been made.

**Line 210: MATLAB should be capitalized**

Noted, change has been made.

**Line 279: Here and throughout the manuscript, capitalize all proper noun geographic locations, e.g., Institute Ice Stream**

Noted, change has been made.

**Line 345: Passive voice: "It is considered. . ."**

Noted, however, no change has been made.

**#Further editing**

**Major Issues**

**1.**

We have added a new method in section 4.4 "Inferring bed elevation using mass conservation (MC) and Kriging" and a new figure 3e which is the modified version of our new bed elevation DEM product using MC at the fast-flowing region of ice and Kriging at the slow-moving ice:

*"Relying on the conservation of mass (MC) to infer the bed between flight lines (Morlighem et al., 2011), we were able to investigate how the bed can be developed further in fast-flowing regions, using a new interpolation technique. To perform the MC procedure, we used InSAR-derived surface velocities from Rignot et al. (2011b), surface mass balance from RACMO 2.3 (Van Wessem et al., 2014), and assumed that the ice thinning/thickening rate and basal melt are negligible. We constrained the optimization with ground penetrating radar from CReSIS, GRADES, IMAFI and SPRI, described above, and used a mesh horizontal resolution of 500 m."*

The purpose of this added product is to demonstrate that there are many ways that our new DEM can be used and modified according to their interpretations of the ice-sheet dynamic in the WS sector region. We have added a new paragraph explaining the details of the new product in section 5.1:

*"The new DEM can be refined further to deal with bumps and irregularities associated with interpolation effects from along-track data in otherwise data sparse regions. We used mass conservation (MC) and kriging to infer the bed elevation (Fig. 3e) beneath the fast and slow-moving ice, respectively, similar to that successfully employed in Greenland (Morlighem et al., 2017). In general, the bed elevation derived from MC and kriging is consistent with our new DEM. However, there are significant changes in the bed morphology beneath fast flowing ice. For example, the tributary of the Foundation Ice Stream has been extended for ~100 km further inland relative to our new DEM."*

**2.**

We have made corrections to account for the way Bedmap2 is projected using the gl04c geoid. (Bedmap2 was produced relative to the gl04c geoid to allow realistic numerical modelling by incorporating the role of relative sea level / glacial isostatic adjustment.) In a previous figure, we did not correct the datum and, so, it was not strictly comparable with our new DEM, because this is produced relative to the WGS 1984 Antarctic Polar Stereographic projection. We have made the necessary corrections by transforming Bedmap2 onto the WGS 1984 Antarctic Polar Stereographic. Consequently, we have reproduced the figure 3c "Difference map" and figure 4. The re-analysis does not have a major difference relative to the previous figures, however. Finally, we have made the necessary adjustments to the figures, and have added text into the methodology section 4.3 as follows:

*"The Bedmap2 bed elevation product (Fretwell et al., 2013) was transformed from the gl04c geoid projection to the WGS 1984 Antarctic Polar Stereographic projection."*

**Minor Issues**

Line 21: "update" to "improve"

Line 23-24: "as BEDMAP2 included only relatively crude ice thickness measurements determined in the field for quality control" to "from the relatively crude measurements determined in the field for quality control purposes used in Bedmap2".

Line 24-25: "This have resulted in the deep parts of the topography not being visible in the fieldwork non-SAR processed radargrams" deleted.

Line 26: "deep trough" to "deep subglacial trough".

Line 28: "at the ice-bed interface" deleted.

Line 29: "sensitivity" to "potential vulnerability".

Line 29: "of the ice sheet" deleted.

Line 32: "in the" to "at".

Line 37: "projected" to "concluded".

Line 41: "in places more than" added.

Line 43: "future" deleted.

Line 44: "project" to "show".

Line 46: "similar manner" to "manner similar".

Line 48-49: "from the parent ice mass" deleted.

Line 49: "together" deleted.

Line 62: "entire" deleted.

Line 64: "late 1960s and.." added.

Line 66: "and" deleted.

Line 66: "thickness" to "ice thickness"

Line 68: "highly" deleted.

Line 74: "line representation" to "lines compared".

Line 74: "observations" to "satellite-derived observations".

Line 75: "limited" to "restricted".

Line 75: "to the community" deleted.

Line 77: "project" to "compilation".

Line 79: "where only 34% of cells have data and 80% have data within the range of 20 km" deleted.

Line 85: "improving knowledge of the land surface beneath the ice sheet. This…" to "…,which…"

Line 87: "the" deleted.

Line 87: "of" to "associated with".

Line 88: "in glaciology" added.

Line 90: "a revised" to "an improved".

Line 96: "The study area is located in" deleted.

Line 96: "covering" to "covers".

Line 97: "The DEM" to "The region covered by the DEM".

Line 98: "with the" to "with".

Line 102: "We used the Differential Interferometry Synthetic Aperture Radar (DInSAR) grounding line (Rignot et al., 2011b) to delimit the ice shelf-facing margin of our grid" added.

Line 102-107: "The data and methods used in this study are mostly similar to Jeofry et al. (2017), however, we have expanded the boundaries of our new bed elevation DEM by focussing across the Weddell Sea sector relative to the Institute ice stream as the main focus in Jeofry et al. (2017). In addition, this paper focuses on discussing the geomorphology and improvement that have been made between our new bed elevation DEM and Bedmap2 product whereas Jeofry et al. (2017) discuss on the hitherto unknown subglacial embayment and its effect to the ice sheet dynamics" deleted.

Line 120: "of the 1970s" deleted.

Line 132: "was" to "were".

Line 133: "was" to "were".

Line 134: "records were" to "record was".

Line 144: "is" to "are".

Line 157: the coordinate for C110 "-82º 37' 30'' N, 280º 59' 13.2''E" has been removed. It has been replaced with "*17 flights were flown from C110, which is located above Institute E2 subglacial lake,…*".

Line 158: the coordinate for Patriot Hills "-80º 19' 60'' N, 278º 34' 60''E" has been removed.

Line 199: "200m" to "200 m".

Line 247-249: "The ice thickness DEM was then subtracted from the ice-sheet surface elevation derived from the combined European Remote Sensing Satellite-1 (ERS-1) radar and Ice, Cloud and land Elevation Satellite (ICESat) laser satellite altimetry DEM (Bamber et al., 2009a), to derive the bed topography" to "The ice thickness DEM was then subtracted from the ice-sheet surface elevation DEM (from European Remote Sensing Satellite-1 (ERS-1) radar and Ice, Cloud and land Elevation Satellite (ICESat) laser satellite altimetry datasets (Bamber et al., 2009a)), to derive the bed topography".

Line 289: "the" deleted.

Line 291: "the" deleted.

Line 298: "It is noted that" deleted.

Line 304: "this is" deleted.

Line 310: "almost" deleted.

Line 312: "a vital constituent" to "an obvious component".

Line 313: "melts" to "allows"

Line 314: "to melt if it is >~2km" added.

Line 314: "on the bed" added.

Line 314: "of the ice" deleted.

Line 315: "At the western margin of the Ellsworth Mountain" to "West of the Ellsworth Mountain".

Line 315: "large" deleted.

Line 317: "In some cases, ice-surface elevation changes have been linked with subglacial lake hydrological change (Siegert et al., 2016a) – the so called 'active' subglacial lakes." Added.

Line 317: "active" added.

Line 318: "The" deleted

Line 319: "The" deleted.

Line 321: "active" added.

Line 323: "East of the Foundation Ice Stream, there are 16 active subglacial lakes distributed along the main trunk of Academy Ice Stream (Wright and Siegert, 2012)" added.

Line 340: "of the" deleted.

Line 343: "and its connection" to "and connects".

Line 350: "It is worth noting that the lake is outside the grid of our new DEM" added.

Line 385: "in the" to "at".

Line 416: "the Institute ice stream, Bungenstock ice rise, Möller and Foundation ice streams" to "the Institute, Möller and Foundation ice streams, and the Bungenstock ice rise".

Line 417: "between the new DEM and that of the previous DEM of the region (Bedmap2)" to "between the new and previous DEMs (i.e., Bedmap2)"

Line 419-421: "the bed elevation of our DEM appears to be slightly lower relative to the bed elevation of Bedmap2 DEM, which is likely the results of the deep sections of the topography not being visible in the fieldwork non-SAR processed radargrams" to "improved processing of existing data has led to some particularly deep regions of bed to be better resolved than in Bedmap2"

Line 422: "that derived from" deleted.

[revised manuscript text omitted]